# A marine cryptochrome with an inverse photo-oligomerization mechanism

Hong Ha Vu [1,7], Heide Behrmann [2,7], Maja Hanić [3], Gayathri Jeyasankar [2], Shruthi Krishnan [1], Dennis Dannecker [2], Constantin Hammer [1], Monika Gunkel [2], Ilia A. Solov'yov [3,4,5], Eva Wolf [1,6] ✉ & Elmar Behrmann [2] ✉

Cryptochromes (CRYs) are a structurally conserved but functionally diverse family of proteins that can confer unique sensory properties to organisms. In the marine bristle worm Platynereis dumerilii, its light receptive cryptochrome L-CRY (*Pd*LCry) allows the animal to discriminate between sunlight and moonlight, an important requirement for synchronizing its lunar cycle-dependent mass spawning. Using cryo-electron microscopy, we show that in the dark, *Pd*LCry adopts a dimer arrangement observed neither in plant nor insect CRYs. Intense illumination disassembles the dimer into monomers. Structural and functional data suggest a mechanistic coupling between the light-sensing flavin adenine dinucleotide chromophore, the dimer interface, and the C-terminal tail helix, with a likely involvement of the phosphate binding loop. Taken together, our work establishes *Pd*LCry as a CRY protein with inverse photo-oligomerization with respect to plant CRYs, and provides molecular insights into how this protein might help discriminating the different light intensities associated with sunlight and moonlight.

Cryptochromes (CRYs) are found in both animals and plants and have evolved from DNA-repairing photolyases (PLs)[1]. Animal CRYs can be subdivided into light-sensitive CRYs, namely type I (Drosophila-like) and type IV CRYs, and light-insensitive CRYs, namely type II (mammalian-like), CRYs[2]. In contrast, plant CRYs appear to be exclusively light-sensitive[3].

While animal and plant CRYs have evolved from different ancestral PL backgrounds[4], they share a highly conserved PL homology region (PHR). Unlike PLs, the CRY PHR is modified by a specific C-terminal expansion (CTE), and sometimes an N-terminal expansion (NTE), postulated to dictate its biochemical function[1,5,6]. The PHR can be structurally sub-divided into two regions, namely an N-terminal α/β-region and a C-terminal α-helical region connected by an extended connector region. The α-helical region contains the binding pocket for the major chromophore, flavin adenine dinucleotide (FAD), which is required for the light response. In the dark resting state of type I CRYs, FAD is non-covalently bound in its oxidized form ($FAD^{OX}$)[7]. Absorption of a blue light photon induces a photoexcited state $FAD^{OX*}$, which in turn abstracts an electron from a nearby tryptophan (TrpH) residue forming a radical pair between a now one-electron reduced anionic semiquinone (ASQ, FAD˙⁻) and a cationic tryptophanyl radical (TrpH˙⁺), which decays back to the ground state within minutes in the dark[7].

The molecular mechanism behind the conversion of redox changes in FAD into an actual physiological signaling state in CRYs is still under debate. Since CRYs have no known enzymatic activity, a signaling state will most likely be defined by one or more

[1]Institute of Molecular Physiology (IMP), Johannes Gutenberg-University Mainz, Hanns-Dieter-Hüsch-Weg 17, 55128 Mainz, Germany. [2]University of Cologne, Faculty of Mathematics and Natural Sciences, Institute of Biochemistry, Zülpicher Straße 47, 50674 Cologne, Germany. [3]Institute of Physics, Carl von Ossietzky University of Oldenburg, Carl-von-Ossietzky Straße 9-11, 26129 Oldenburg, Germany. [4]Research Center for Neurosensory Sciences, Carl von Ossietzky University of Oldenburg, Carl-von-Ossietzky Straße 9-11, 26111 Oldenburg, Germany. [5]Center for Nanoscale Dynamics (CENAD), Carl von Ossietzky Universität Oldenburg, Ammerländer Heerstr. 114-118, 26129 Oldenburg, Germany. [6]Institute of Molecular Biology (IMB), 55128 Mainz, Germany. [7]These authors contributed equally: Hong Ha Vu, Heide Behrmann. ✉e-mail: evawolf1@uni-mainz.de; elmar.behrmann@uni-koeln.de

conformational changes. The highly divergent CTE is a likely candidate for these[1,4,6]. Proteolytic experiments have suggested a conformational change involving the CTE upon blue light excitation for type I[8–10], type IV[11], and plant CRYs[12]. For the *Drosophila melanogaster* CRY (*Dm*Cry) this has been corroborated by time-resolved x-ray solution scattering experiments and molecular dynamics (MD) simulations[13]. Indeed, due to its unusually short CTE, forming only a short C-terminal tail helix (CTT, helix α23), *Dm*Cry has been a focal point of attention in attempts to unravel the molecular coupling between FAD photoreduction and CTE conformational changes. FAD is not known to undergo a conformational change during photoactivation, but the photoinduced electron transfer results in a net negative charge at the flavin isoalloxazine ring. Structural data have revealed that *Dm*Cry His378 is positioned between the isoalloxazine ring and a so-called FFW motif at the C-terminal end of the CTT[9,14], similar to an active site histidine that is essential for PL DNA repair[15]. MD simulations suggest that the hydrogen-bonding network connecting His378, isoalloxazine ring and FFW motif is indeed sensitive to an additional negative charge on the isoalloxazine ring[16], which was later confirmed by further experimental data[17]. The recent surge in structural data of CRYs from other species has suggested that this proposed coupling mechanism may be conserved in the CRY superfamily, albeit with lineage-specific fine-tuning, particularly for plant CRYs, since His378 is not conserved in the plant lineage and their most stable FAD form is a one-electron reduced neutral semiquinone (NSQ, FADH')[7]. Interestingly, while it is widely accepted that blue light-activated plant CRYs oligomerize and that this is important for downstream signaling[18], signaling by non-plant CRYs is largely thought to depend on the conformation of the CTE in the CRY monomer[19]. There are currently two known exceptions to this dogma: Firstly, dark state dimers have been reported for the animal-like CRY from the algae *Chlamydomonas reinhardtii* (*Cra*Cry)[20], although this could not be reproduced in later experiments[21]. Secondly, for the photoactive L-CRY from the bristle worm *Platynereis dumerilii* (*Pd*LCry), dimers were observed in in vitro assays using SEC-MALS and proposed to be of physiological importance[22].

*Pd*LCry is one of two light-sensitive CRYs identified in this marine invertebrate, which is an important model system for both genetic and molecular studies[23]. Genetic and behavioral experiments have identified *Pd*LCry as an important timekeeper of its lunar-circle dependent spawning behavior[22,24]. Both in vivo and in vitro data revealed a peculiar light-sensitivity that allows *Pd*LCry to respond to dim moonlight[22,24].

Here, we solve the structure of both dark state and blue light-illuminated *Pd*LCry using cryo-electron microscopy (cryo-EM) in combination with single-particle reconstruction techniques. Our dark state structure reveals a distinct dimerization interface centered on helix α8, while the blue light-illuminated structure shows that *Pd*LCry monomerizes upon intense illumination. Furthermore, our structural analysis together with functional data suggest the phosphate binding loop (PBL) as a central element for the molecular coupling between the chromophore, the CTT and the dimer interface. Together, our study uncovers a mechanism of light-induced inverse oligomerization, which has important implications for our understanding of the ability of *Pd*LCry to discriminate moon- from sunlight.

## Results

### *Pd*LCry features subtle but distinct sequence differences separating it from classical type I CRYs

As a first step to decipher why *Pd*LCry is able to dimerize, we performed an evolutionary analysis of its sequence. This analysis confirmed the previous association of *Pd*LCry with type I CRYs[25]. However, we find that *Pd*LCry is distinctly separated from classical members of this lineage (Fig. 1a). Indeed, a detailed alignment of *Pd*LCry with the only type I CRY with a known structure to date, *Dm*Cry[9,14], reveals subtle but distinct sequence differences in regions reported to be important for the molecular properties of *Dm*Cry: *Pd*LCry has a shortened protrusion loop, an FCW sequence instead of an FFW sequence at the CTT, and a tyrosine in the sulfur loop (Fig.1b, Supplementary Fig. 1, Supplementary Table 1). However, even a close comparison of their sequences did not provide a molecular explanation for the different oligomeric states of the two proteins. Therefore, we set out to elucidate the molecular architecture of *Pd*LCry using molecular cryo-electron microscopy (cryo-EM).

### *Pd*LCry forms a dimer in the dark state featuring a distinct dimer interface

*Pd*LCry was heterologously expressed in *Sf*9 insect cells and purified under far-red light conditions (Supplementary Fig. 6a) to ensure that

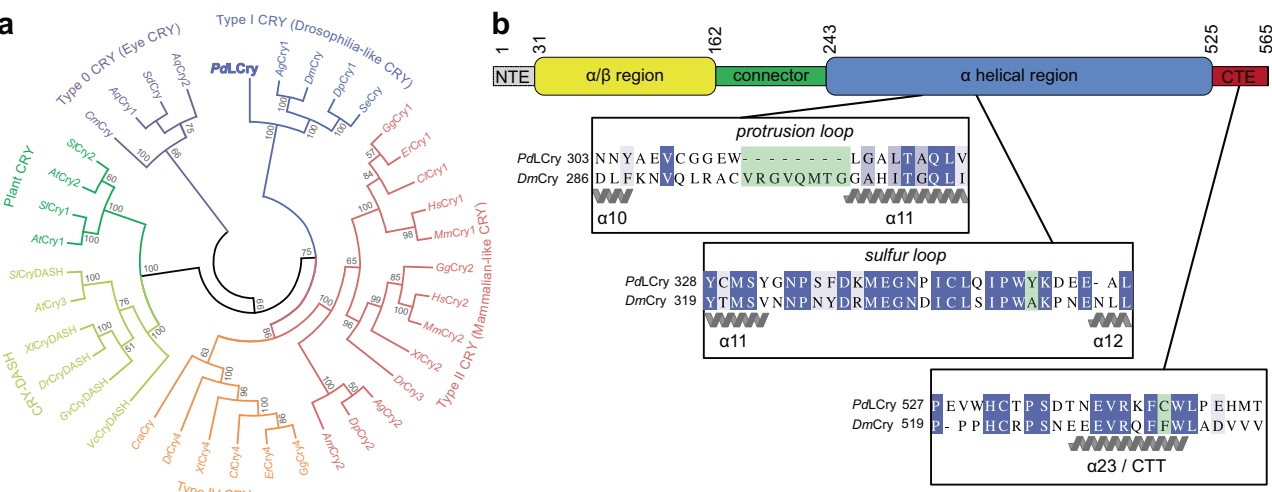

**Fig. 1 | *Pd*LCry belongs to the type I CRY lineage but has unique sequence features. a** Evolutionary analysis of the CRY family using a maximum likelihood approach shows that *Pd*LCry belongs to the type I CRY lineage (blue), but forms a subbranch of this lineage. Small numbers indicate the bootstrap support of the branch association. Full species names and protein sequences used are listed in Supplementary Table 1. **b** Cartoon representation of the domain composition of *Pd*LCry. Small numbers indicate the residue at the region boundary. Inserts show excerpts from a sequence alignment to *Dm*Cry, highlighting differences (marked green) in the protrusion loop, the sulfur loop and the C-terminal tail helix (CTT) FFW motif. See Supplementary Fig. 1 for the complete sequence alignment. Secondary structure annotation (α-helix indicated in gray) has been published in[9].

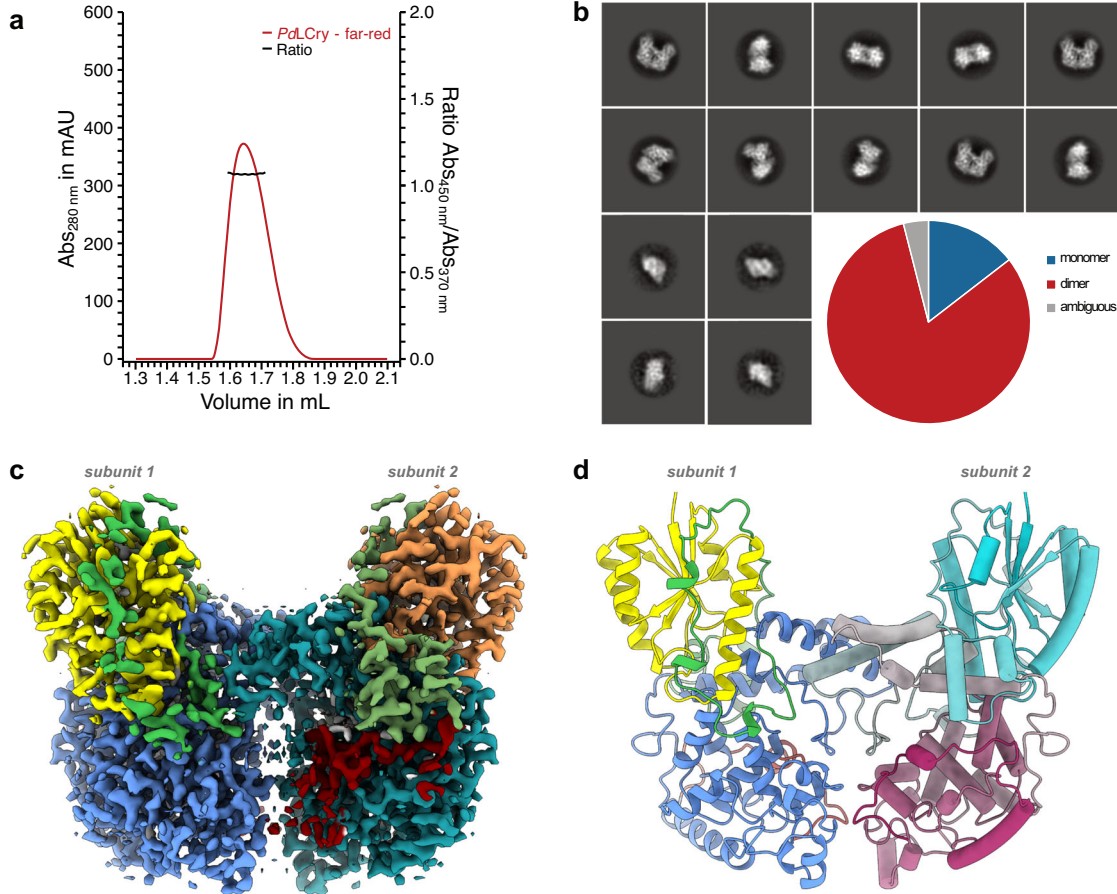

**Fig. 2 | FAD<sup>OX</sup>-bound *Pd*LCry forms a unique dimer in the dark state. a** Analytical size exclusion chromatography of *Pd*LCry using a Superdex 200 Increase 5/150 GL column under far-red light conditions shows a slightly tailing elution peak following absorption at 280 nm (red curve), likely corresponding to a dimeric state of the protein. The nearly equal absorbance at 370 nm and 450 nm (black line; $Abs_{450nm}$/$Abs_{370nm}$ = 1.06) confirms that *Pd*LCry was successfully maintained in the FAD<sup>OX</sup> bound dark state[26]. **b** Representative 2D class averages of an unsupervised 2D classification of the dark state *Pd*LCry cryo-EM sample show predominantly dimeric particles, with few monomeric particles. Quantification of all protein-shaped particles suggests that approximately 80% of the protein is present as a dimer after

vitrification under far-red light conditions. **c** Surface representation of the cryo-EM density map of *Pd*LCry shows a butterfly-shaped dimer, with a large interaction interface formed by the α-helical region (blue) with contributions from the connector region (green). Subunit 2 is shown in darker shades. In the shown orientation, the CTE is only visible for subunit 2 (dark red). Scale bar corresponds to 10 Å. **d** Atomic model built into the cryo-EM density map shows that *Pd*LCry adopts the typical PL/CRY fold and that both subunits are arranged in parallel. Subunit 1 is shown in cartoon style using a coloring scheme identical to that used for subunit 1 in panel (**c**). Subunit 2 is shown in a pipes-and-planks style using a color gradient from cyan (N-terminus) to maroon (C-terminus).

FAD remained in its oxidized ground state (FAD<sup>OX</sup>) as previously described[22]. As for Poehn et al., *Pd*LCry purified as a homodimer under these conditions with an absorbance ratio of 450 nm to 370 nm of 1.06 (Fig. 2a), indicating an FAD<sup>OX</sup> loaded protein[26]. The purified protein was vitrified in liquid ethane under far-red light conditions to produce cryo-EM grids of the dark state. Data acquisition and unsupervised 2D classification indeed revealed predominantly dimeric particles with a small fraction of putative monomers (Fig. 2b). We were able to refine the subset of particles composed of two subunits to obtain a dimeric structure with the *Pd*LCry subunits arranged in parallel. As we did not observe any asymmetry during the reconstruction procedure, we enforced C2 symmetry in later reconstruction stages, yielding a final cryo-EM density map at 2.6 Å resolution (Fig. 2c, Supplementary Fig. 2). The exquisite resolution of the final, C2 symmetric 3D map allowed us to build an almost complete atomic model of *Pd*LCry (Fig. 2d, Supplementary Table 2). We observe a near stochiometric occupancy of the ligand FAD, but no density for additional antenna chromophores or other nucleotides. Our map shows an overall fold reminiscent of that observed in other type I CRYs, with an N-terminal α/β region (31-161) and a C-terminal α-helical region (243–524) connected by a largely well-resolved connector (162-242). While we do not observe density for

the *Pd*LCry specific NTE (1–30), the *Pd*LCry specific CTE (526-565) is largely well resolved and features a well-resolved C-terminal tail helix (CTT, 537-544), which resembles the helix observed in *Dm*Cry[9,14]. The density for the last nine C-terminal residues (557-565) was strongly fragmented, suggesting considerable flexibility. The arrangement of the subunits in the dimer does not resemble either that of plant CRYs, nor that of the crystal dimer arrangement of *Dm*Cry (Supplementary Fig. 3a, b). Instead, the dimer interface in *Pd*LCry is predominantly formed by the N-terminal part of the α-helical region with some contribution from the connector region. Molecular dynamics simulations show that the solvent-accessible surface area buried in the interface is comparable to that of the biologically relevant *At*Cry2 dimer, despite its different arrangement (Supplementary Fig. 3c–e).

The main dimer interface is formed between helices α8 of both subunits, with helices α8 running antiparallel at an angle of 42 degrees, flanked by interactions mediated by the connector region and the PBL (Fig. 3a, b). While helix α8 and the flanking regions adopt a fold that overlays well with available structures of *Dm*Cry, the archetypical type I CRY, several key residues differ (Fig. 3c): the positions of Thr253 and Thr260 in *Pd*LCry are occupied by bulkier amino acids in *Dm*Cry, namely Glu236 and His243, which would likely clash in a similar dimer

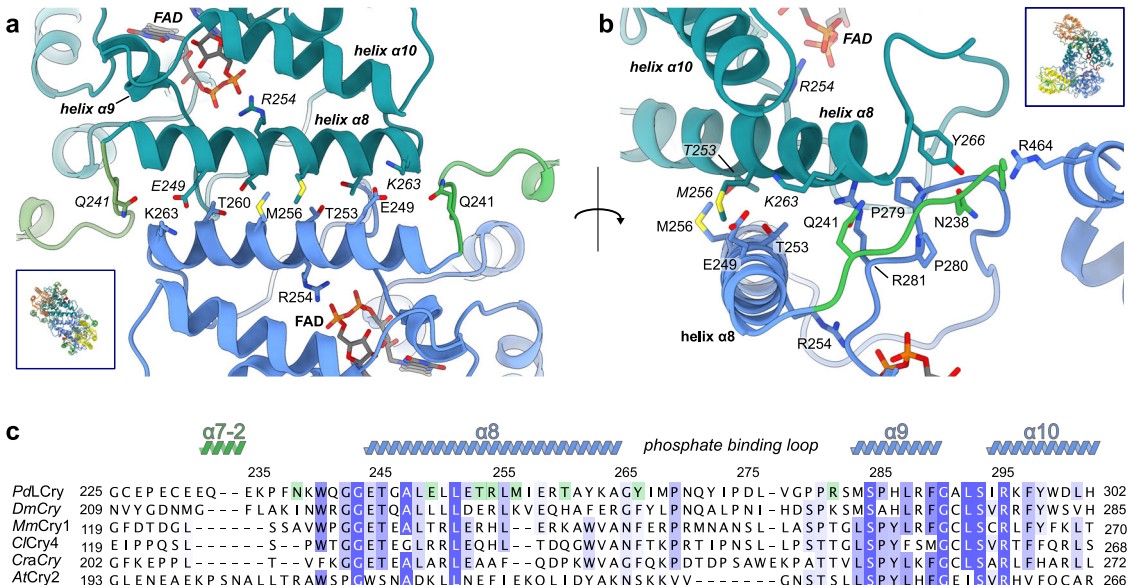

**Fig. 3 | Helix α8 and its flanking regions are central to the dimer interface.** a Close-up view of the dimer interface and **b** 90-degree rotated view highlight the importance of helix α8 for the dark state dimer. Helix α8 of subunit 1 (using the same color code as in Fig. 2) runs antiparallel to helix α8 of subunit 2 (colored in darker shades as in Fig. 2), at an angle of 42 degrees. The contact point between the two helices is at the central Met256, flanked by Thr253 and Thr260. Glu249 and Arg281 appear to lock the helices in place, with possible additional stabilization by interactions centered around Tyr266 of the phosphate binding loop. Arg254 of

helix α8 contacts the FAD chromophore. Inserts show the respective orientation of the dimer in (**a**) and (**b**). **c** Sequence alignment of *Pd*LCry with plant, animal and animal-like CRYs of known structure indicates that the observed dimer interface is unique to *Pd*LCry. Residues discussed in the text are highlighted in green. Secondary structure elements and amino acid numbering of *Pd*LCry are shown above the sequences. Full species names and protein sequences used are listed in Supplementary Table 1.

arrangement. Similarly, the position of the central Met256 is occupied by a bulkier lysine (Lys239), and the position of Glu249 is occupied by a non-polar leucine residue (Leu232) in *Dm*Cry. Apart from the differences in helix α8, we note unique dimer interactions involving Tyr266 of the PBL, especially with Pro279 (PBL) and Asn238 (connector) from the other subunit. These interactions are not possible in *Dm*Cry because Tyr266 is replaced by a phenylalanine (Phe249), which appears to be conserved in other animal and animal-like CRYs (Fig. 3c). In addition, the position of Asn238 is occupied by a non-polar isoleucine (Ile221) in *Dm*Cry. Indeed, MD simulations confirm the importance of Asn238, Glu249, and Tyr266 as key residues that, due to their polar side chains, stabilize the dimer interface through hydrogen bond formation. (Supplementary Fig. 3f).

**Structural elements highlight a molecular connection between the chromophore FAD, the CTT, and the dimer interface**
A chain of four tryptophan residues, comprising Trp$_A$428, Trp$_B$405, Trp$_C$351, and Trp$_D$402, lines the path from the FAD to the protein surface and occupies identical positions as observed in the archetypical type I CRY *Dm*Cry (Fig. 4a, b). However, the sulfur loop of *Pd*LCry has an additional surface-exposed Tyr352 that occupies a position similar to the terminal tyrosine in the type IV CRY *Cl*Cry4[11] (Fig. 4b). In *Cl*Cry4, this terminal tyrosine (Tyr319) has been shown to increase the lifetime of the radical state[11], as also observed for the terminal tyrosine residue that replaces Trp$_D$ in *Cra*Cry[20]. To test whether this terminal tyrosine does indeed affect the efficiency of FAD photoreduction in *Pd*LCry, we replaced Tyr352 with either an alanine or an aspartate as found in *Dm*Cry and *Cra*Cry (Fig. 4b, Supplementary Fig. 1). Both *Pd*LCry$^{Y352A}$ and *Pd*LCry$^{Y352D}$ form dimers in the dark (Supplementary Fig. 6i, j), but both mutations lead to a less efficient FAD photoreduction under illumination conditions mimicking naturalistic sunlight conditions at the habitat of *P. dumerilii*[22]. Indeed, for both mutations photoreduction was slower and incomplete after 20 min sunlight illumination (Supplementary Fig 4a, b), different from

wildtype *Pd*LCry that is almost completely photoreduced using identical conditions (Supplementary Fig. 4c).

In the dark state, the CTE is anchored in the vicinity of the FAD binding site by a CTT comprising residues 537 to 545. This CTT occupies an identical PL-DNA substrate binding pocket as in *Dm*Cry[9] (Fig. 4c). Interestingly, Cys544 of the FCW motif in *Pd*LCry adopts a conformation similar to Phe535 of the FFW motif in *Dm*Cry and, despite being less bulky, binds into an almost identical hydrophobic pocket delineated by Val431, Leu440, Val540 and Phe446, the latter occupying a similar position to *Dm*Cry Val437 (Fig. 4c). Notably, the smaller size of the cysteine side chain compared to the phenylalanine side chain allows His390 of helix α14 to adopt a different conformation compared to the corresponding Asn382 in *Dm*Cry (Supplementary Fig. 4d). Avian type IV CRYs[11] and the animal-like *Cra*Cry[27] also have a histidine at this position instead of an asparagine, but in a rotamer superimposed on Asn382 in *Dm*Cry (Supplementary Fig. 4d). Compared to *Pd*LCry, this different histidine rotamer could possibly be due to the absence of a CTE in the truncated constructs used for crystallography.

As for *Dm*Cry, Trp545 in *Pd*LCry is positioned at the end of the CTT. His386 (corresponding to *Dm*Cry His378) and the PBL residues Pro274 (corresponding to *Dm*Cry Pro257) and Gln271 delineate the binding pocket of Trp545. Thereby, the FCW motif in *Pd*LCry is coupled to the dimer interface through the PBL. Interestingly, Gln271 is residing in a similar position as *Dm*Cry Arg298 from the protrusion loop, which is considerably shorter in *Pd*LCry compared to *Dm*Cry (Fig. 4c). In *Dm*Cry Arg298 has been implicated in stabilizing the dark state by anchoring *Dm*Cry Trp536[9,28], suggesting that Gln271 could play a similar role in *Pd*LCry.

Phe543 occupies an identical position to *Dm*Cry Phe534 and is similarly delineated by Trp430 (*Dm*Cry Trp422) and His386 (*Dm*Cry His378). Unlike in *Dm*Cry, the CTE is additionally stabilized by a cation-π interaction between Trp530 from the CTE and Arg174 from the connector region (Supplementary Fig. 4e), which is neither possible in

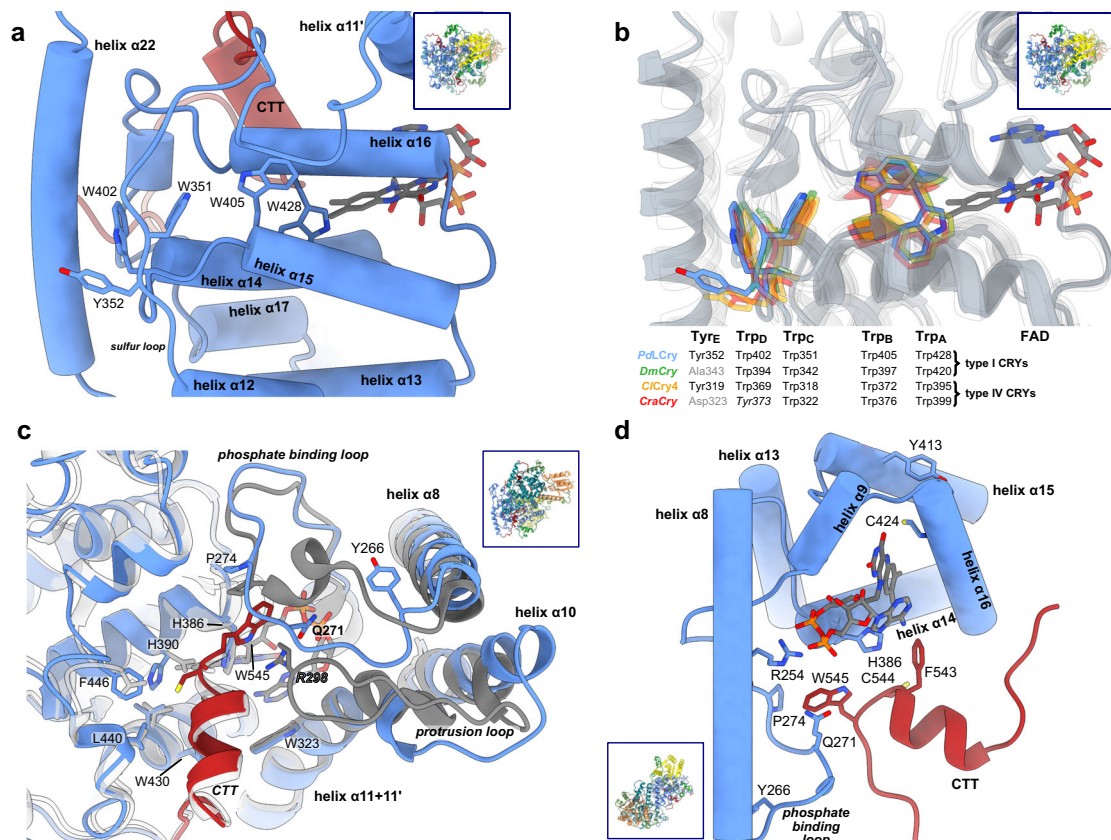

**Fig. 4 | Structural data suggest that the PBL links chromophore activation, dimerization and the CTT release. a** *Pd*LCry has five aromatic residues potentially involved in the photoinduced electron transfer, namely Trp428 (Trp$_A$), Trp405 (Trp$_B$), Trp351 (Trp$_C$), Trp402 (Trp$_D$) and Tyr352 (Tyr$_E$). **b** The core Trp-triad consisting of Trp$_A$, Trp$_B$, and Trp$_C$ is conserved in the light-responsive type I and IV CRYs whose structure has been solved to date: *Pd*LCry (blue), *Dm*Cry (green; PDB 4JZY), *Cl*Cry4 (orange; PDB 6PU0), and *Cra*Cry (red; PDB 6ZM0). The fourth aromatic residue, which extends this to a Trp-tetrad, is conserved except for *Cra*Cry, which features a tyrosine at this position. Only *Pd*LCry and the type IV CRY *Cl*Cry4 further extend the electron transport chain by a fifth member, which is a tyrosine in both proteins. *Pd*LCry is shown in transparent gray, the other CRY structures are shown in transparent white. **c** An overlay of the *Pd*LCry (colored) and the *Dm*Cry (gray and white; PDB 6WTB) structures shows that both share a CTT that is anchored in the DNA substrate binding pocket of PLs. However, instead of an FFW motif, the *Pd*LCry CTT is anchored by an FCW motif. The smaller Cys544 interacts

with His390, which occupies a different position compared to *Dm*Cry Asn382. Importantly, the *Pd*LCry protrusion loop is considerably shorter than its wild-type *Dm*Cry counterpart (dark gray). This results in a different contact to the FCW motif: whereas in *Dm*Cry Arg298 (gray) from the protrusion loop contacts the CTT tryptophan residue, in *Pd*LCry this stabilizing contact is mediated by Gln271 from the phosphate binding loop, which also harbors Tyr266. Therefore, in *Pd*LCry the CTT is structurally linked with the dimer interface helix α8, rather than with the helices α10 and α11 as observed in *Dm*Cry. **d** FAD is bound in a typical type I CRY FAD-binding site, with Cys424 contacting the N5 position of the isoalloxazine ring of FAD, and Tyr413 blocking solvent access to FAD. Our dark state structure reveals a molecular network surrounding FAD and connecting it to both the CTT and the dimer interface helix α8. His386 is positioned between FAD and the FCW motif, while Gln271 from the phosphate binding loop connects helix α8 to Trp545 of the CTT-anchoring FCW motif, and Arg254 connects this helix to the chromophore. Inserts show the orientation of the dimer for each panel.

---

*Dm*Cry nor *Cl*Cry4 as the corresponding residues are Pro521 and Ile153, and Arg497 and Ser146 respectively. C-terminal to the CTT, the remaining CTE passes through a cleft defined by both subunits of the dimer, although the more fragmented density in this region implies conformational flexibility.

The FAD chromophore bound adjacent to the CTT is stabilized by a binding pocket defined primarily by residues from helices α9, α14, and α16, the PBL and the α15-α16 loop (Fig. 4c, d). As in other type I CRYs, the N5-interacting residue is a cysteine, namely Cys424 (*Dm*Cry Cys416), which has been proposed to disfavor the conversion to the two-electron reduced NSQ state of the chromophore FAD[9,14,29]. Solvent access to the FAD binding site[7] could be blocked by Tyr413, which occupies an equivalent position as Leu405 in *Dm*Cry. As reported for *Dm*Cry[16,17], His386 (*Dm*Cry His378) bridges the FAD chromophore to the FCW motif of the CTT and is thus in a suitable position to allow the electronic state of the isoalloxazine ring to be transmitted to the CTE.

Interestingly, in addition to the Gln271-mediated coupling between helix α8 and the CTT, the dimer interface helix α8 is also directly coupled to the FAD via Arg254, which is located at the center

of helix α8 (Fig. 4d). We therefore asked ourselves whether there is a mechanistic coupling between the redox state of FAD, the CTE and the oligomeric state of *Pd*LCry.

## Activation of *Pd*LCry by intense blue light disrupts the dimer

To test whether *Pd*LCry does indeed respond to blue light by changing its oligomeric state, we performed size-exclusion chromatography (SEC) under continuous and intense blue light irradiation. In comparison with the elution pattern observed under far-red light conditions, we observe a shifted elution peak, likely corresponding to monomeric *Pd*LCry, with an absorbance ratio of 450 nm to 370 nm of 0.28, suggesting an ASQ-loaded protein (Fig. 5a). *Pd*LCry had previously been shown to revert to the ground state within minutes[22]. Therefore, cryo-EM samples were vitrified in liquid ethane within less than 5 s after intense blue light irradiation. Data acquisition and unsupervised 2D classification using identical conditions as for the dark state dataset indeed revealed almost exclusively monomeric particles (Fig. 5b). Unfortunately, the low molecular mass combined with a strong preferential orientation of the particles on the grid precluded us from

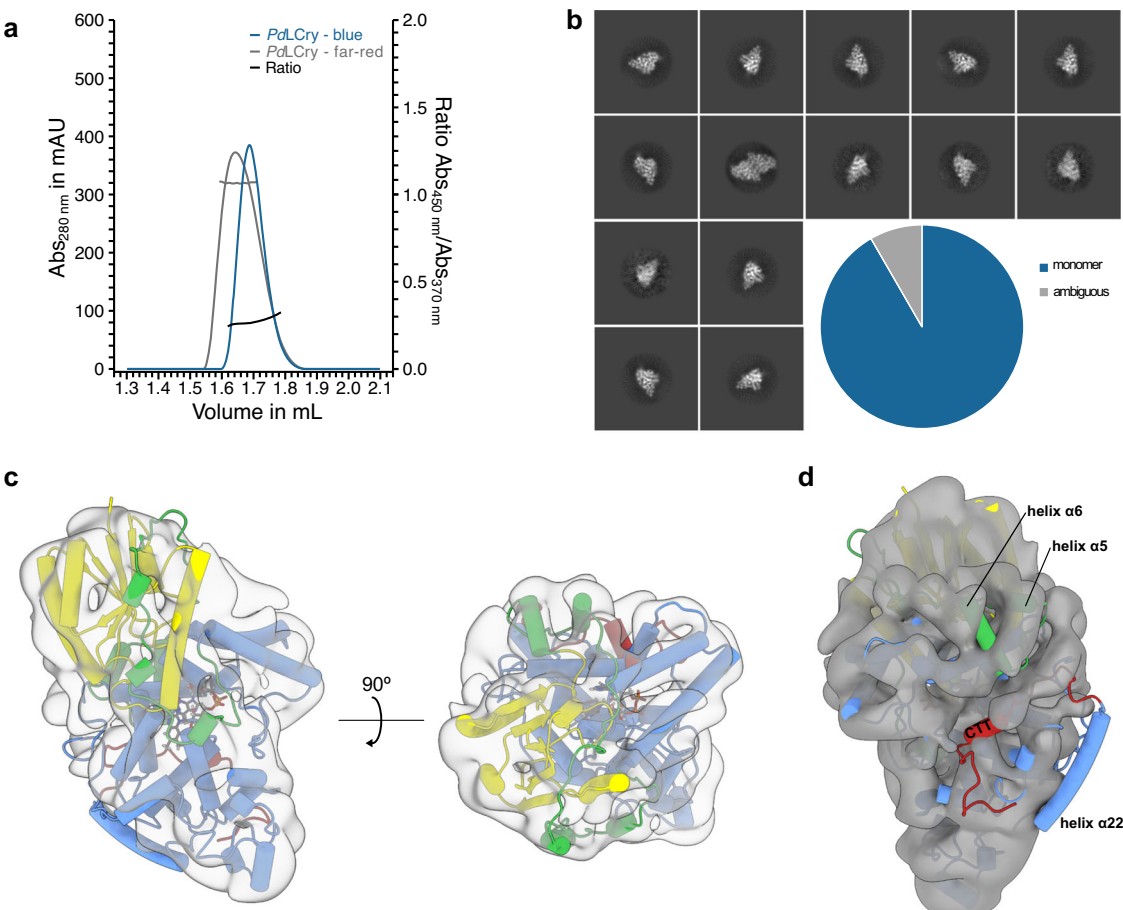

**Fig. 5 | Under intense illumination FAD^ASQ-bound *Pd*LCry disassembles into a monomer. a** Analytical size exclusion chromatography using a Superdex 200 Increase 5/150 GL column of *Pd*LCry under continuous blue light conditions shows an elution peak following absorption at 280 nm (blue curve), which, based on comparison with the elution profile of the dark state SEC (gray curve), likely corresponds to a monomeric state of the protein. The dominant absorption at 370 nm compared to 450 nm (black line; $Abs_{450nm}/Abs_{370nm} = 0.28$) suggests that *Pd*LCry was maintained in the FAD^ASQ-bound photoactivated state, in which the anionic semiquinone radical shifts the spectral properties of FAD[26]. **b** Representative 2D class averages of an unsupervised 2D classification of the photoactivated *Pd*LCry cryo-EM sample confirm the absence of dimeric particles after illumination.

Quantification of all protein-shaped particles implies that at least 90% of the protein is monomeric. Only a few classes are ambiguous, showing two subunits but in an unusual arrangement, possibly due to the close packing of the particles on the cryo-EM grid. **c** Surface representation of the cryo-EM density map of photoactivated *Pd*LCry low-pass filtered to 8 Å shows a classical PL/CRY-shaped monomer. Due to the limited data quality, we refrained from building an atomic model. However, a single subunit of the dark state *Pd*LCry structure (using the same colors as in Fig. 2) fits well with the observed density. **d** Apart from helix α22, for which there is no clear density, parts of the connector region, especially helices α5 and α6, appear to adopt a different conformation, and the density for the CTT and C-terminal parts of the C-terminal expansion is largely fragmented. Scale bar corresponds to 10 Å.

obtaining a high-resolution reconstruction (Supplementary Fig. 5). Due to the limited resolution and strong anisotropy of the 3D cryo-EM map, we refrained from building an atomic model for the activated state. Nevertheless, rigid body fitting of a single subunit from our dark state atomic model into the monomeric 3D map reveals a general fit (Fig. 5c), however with poorer fits for helix α22 and the CTT (Fig. 5d). Both elements have previously been suggested to become more mobile upon blue light activation for other CRY proteins[13,21].

**Mutagenesis highlights the role of helix α8, the PBL and the CTT in *Pd*LCry dimer formation**

To identify the structural elements that control the oligomeric state of *Pd*LCry, we introduced selected amino acid mutations and investigated the oligomeric state of the mutant proteins by performing SEC under far-red light conditions. First, we investigated direct mutations of the dimer interface helix α8. To study the effect of steric clashes, we introduced a bulky side chain into helix α8 by replacing Thr253 of helix α8 with an arginine. Indeed, this shifts the elution volume of *Pd*LCry^T253R towards lower mass (Fig. 6a, Supplementary Fig. 6b). Similarly, introducing a charge reversal by replacing Glu249 of helix α8

with an arginine shifts the elution volume of *Pd*LCry^E249R towards lower mass (Fig. 6a, Supplementary Fig. 6c). Combining both T253R and E249R as a double mutation in *Pd*LCry^E249R,T253R shifts the SEC elution volume to the same volume as observed for the *Pd*LCry^E249R and *Pd*LCry^T253R proteins (Fig. 7a, Supplementary Fig. 6d). Comparing the elution volumes of *Pd*LCry^T253R, *Pd*LCry^E249R and *Pd*LCry^E249R,T253R with the elution profile of wildtype *Pd*LCry, we note that the elution peak of the mutations coincides with a shoulder in the elution profile of wildtype *Pd*LCry. Since we observed a subpopulation of monomeric particles in the dark state cryo-EM dataset (Fig. 2b), we assume that this shoulder peak corresponds to these monomeric particles, further corroborating that the direct interface mutations lead to a monomerization even in the dark state.

Next, based on the hypothesis that CTT undocking is a downstream effect of photoexcitation in light-sensitive CRY proteins, we investigated whether there is a functional coupling between the CTT and the oligomeric state in *Pd*LCry. To this end, we individually mutated Phe543 and Trp545 of the FCW motif, which anchor the CTT to the PL-DNA substrate binding pocket, to alanine (Supplementary Fig. 6e, f). Indeed, both *Pd*LCry^F543A and *Pd*LCry^W545A elute at volumes

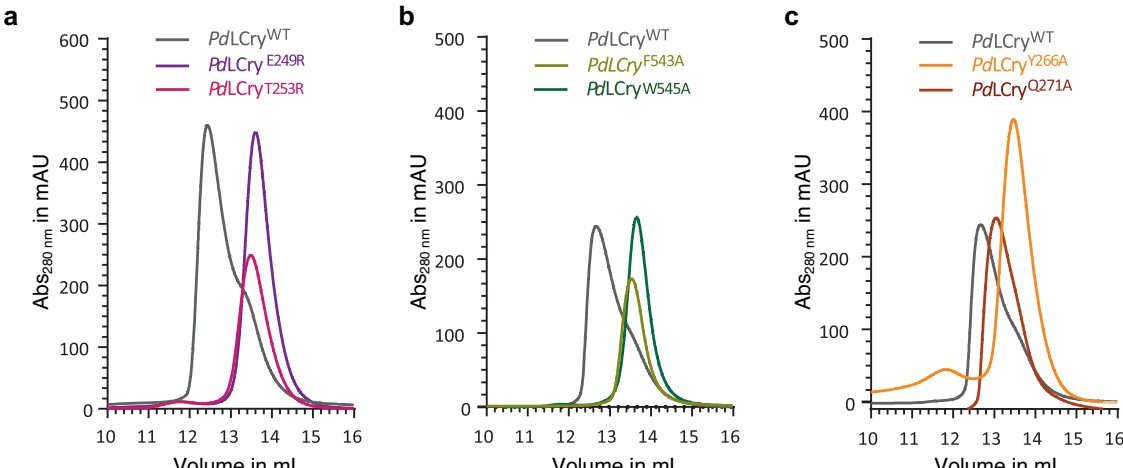

**Fig. 6 | Helix α8, the PBL, and the CTT are involved in *Pd*LCry dimer formation. a** Analytical size exclusion chromatography of wildtype (wt) *Pd*LCry (gray) using a Superdex 200 10/300 GL column under far-red light conditions shows a main elution peak at 12.5 mL with a significant shoulder at 13.3 mL. This shoulder suggests a partial dimer dissociation under far-red light conditions as also observed during cryo-EM (compare Fig. 2b). Using the same SEC conditions, *Pd*LCry^E249R (purple) and *Pd*LCry^T253R (pink), with single mutations in the dimer helix α8, show a main elution peak coinciding with the shoulder of wildtype *Pd*LCry (gray), suggesting that these mutations disrupt the dimer. **b** Analytical size exclusion chromatography of *Pd*LCry^F543A (light green) and *Pd*LCry^W545A (dark green) under far-red light conditions, with single mutations in the FCW motif anchoring the CTT, show a main elution peak coinciding with the shoulder of wildtype *Pd*LCry, suggesting that uncoupling the CTT from its binding pocket disrupts the dimer. **c** Analytical size exclusion chromatography of *Pd*LCry^Y266A (orange) and *Pd*LCry^Q271A (brown) under far-red light conditions, with single mutations in the PBL, also shows an effect on the elution profile. *Pd*LCry^Y266A (orange) shows a main elution peak coinciding with the shoulder of wildtype *Pd*LCry, consistent with a disruption of the dimer. *Pd*LCry^Q271A (brown) shows an elution profile with the main peak shifted to 13 mL and a shoulder at 13.5 mL, which could suggest a partial disruption of the dimer. See Supplementary Fig. S6 for final SEC purification step of wildtype and mutant *Pd*LCry proteins with SDS-PAGE quality control.

implying a monomer (Fig. 6b), suggesting a crosstalk of the CTT conformation to the dimer interface in *Pd*LCry.

Our cryo-EM-derived atomic model highlights the PBL as an important structural element that is directly involved in dimer formation, via hydrogen bonds mediated by Tyr266, and connects the dimer interface to the CTT, via an interaction between Gln271 and Trp545 (Fig. 4c, d). While *Pd*LCry^Y266A, in which Tyr266 of the PBL has been replaced by an alanine that cannot participate in side-chain hydrogen bonding, elutes with a volume clearly indicative of a monomer, the mutation of Gln271 to alanine has a less clear effect, and *Pd*LCry^Q271 shows an elution volume between that expected for dimer and monomer (Fig. 6c, Supplementary Fig. 6g, h).

### The oligomeric state of *Pd*LCry affects the kinetics of dark recovery

Given the unique sensitivity of *Pd*LCry compared to the archetypal monomeric type I CRY *Dm*Cry[22,24], and the observation that *Pd*LCry responds to intense illumination with monomerization, we asked whether the oligomeric state of *Pd*LCry has an influence on the properties of its photoreaction. To ensure constitutive monomer formation throughout the spectroscopic measurements, we used the combined dimer interface mutation *Pd*LCry^E249R,T253R for these experiments. Based on an absorbance ratio of 450 nm to 370 nm of 1.14, which is comparable to that of the wildtype protein, *Pd*LCry^E249R,T253R eluted as a monomer loaded with FAD^OX under far-red light conditions (Fig. 7a). Therefore, the double mutation does not appear to affect the ground state of the FAD chromophore. Under continuous and intense blue light illumination, both wildtype *Pd*LCry and *Pd*LCry^E249R,T253R elute with almost identical volumes, further corroborating that the double mutation is indeed a monomer. Under these conditions, the absorbance ratio of 450 nm to 370 nm drops to 0.28 (Fig. 7b) for both wildtype *Pd*LCry and *Pd*LCry^E249R,T253R, suggesting that while the double mutation disrupts the dimer, it does not affect the overall ability of the chromophore in *Pd*LCry to be photoreduced to the FAD^ASQ-state by blue light photons.

To analyze whether the oligomeric state affects the kinetics of *Pd*LCry photoreduction or dark recovery, we performed time-resolved UV/Vis spectroscopy. UV/Vis spectroscopic analyses confirmed that in monomeric *Pd*LCry^E249R,T253R FAD is completely photoreduced to FAD^ASQ after 2 min of illumination with intense blue light (41 W m^{-2} at the sample), as also observed for wildtype *Pd*LCry (Supplementary Fig. 7a). To gain insight into the photoactivation of monomeric *Pd*LCry under sunlight conditions, we also analyzed FAD photoreduction upon illumination with naturalistic sunlight[22], which has a ~ 9-fold lower intensity (4.6 W m^{-2} at sample) than our blue light source. We found that monomeric *Pd*LCry^E249R,T253R - like wildtype *Pd*LCry - showed slower FAD photoreduction kinetics under sunlight compared to blue light, reaching the sunlight state with fully photoreduced FAD^ASQ within approximately 20 min (Fig. 7c, d). Moreover, the monomeric mutant showed a wildtype-like sunlight photoreduction rate (average $t_{1/2}$ wildtype = 22.7 ± 0.4 s; $t_{1/2}$ mutant = 22.6 ± 0.8 s). Thus, dimer formation is not essential for efficient FAD photoreduction under naturalistic sunlight conditions. This would allow sunlight photoreduction to continue to completion, even if sunlight-induced dimer dissociation should occur early in the photoreduction process. However, wildtype *Pd*LCry had a slightly slower dark recovery rate than monomeric *Pd*LCry^E249R,T253R (average $t_{1/2}$ wildtype = 185.2 ± 8.8 s; $t_{1/2}$ mutant = 156.7 ± 1.1 s) (Fig. 7e, f).

### Discussion

Our work confirms that *Pd*LCry from the marine bristle worm *Platynereis dumerilii* is indeed able to form dimers, as previously proposed[22,24], and reveals a dimerization interface centered on helix α8 (Fig. 3), which has not been observed in CRY proteins before (Supplementary Fig. 3). While all CRY and PL proteins structurally characterized so far share the same fold and a high degree of structural similarity, it has been shown for plant CRYs that oligomerization interfaces can depend on few surface residues[30], complicating predictions solely based on CRY sequences. For *Pd*LCry, we could show that dimerization depends on the presence of specific amino acids

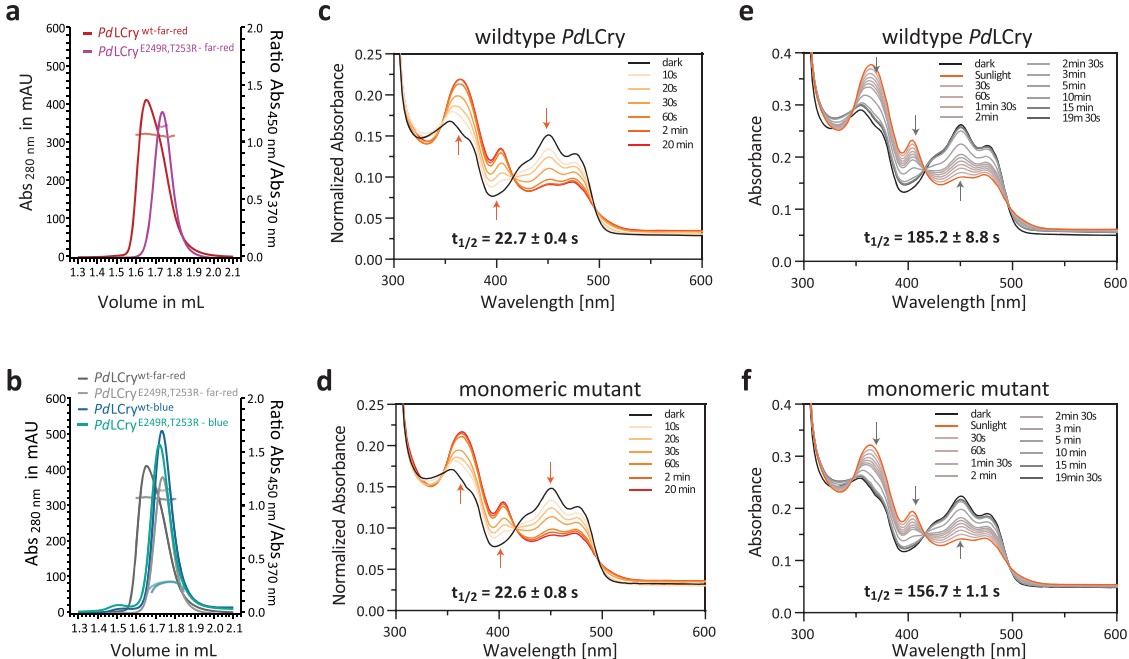

**Fig. 7 | The oligomeric state of *Pd*LCry affects its dark recovery kinetics.**
**a** Analytical size exclusion chromatography (SEC) using a Superdex 200 Increase 5/150 GL column of *Pd*LCry (red) and *Pd*LCry^E249R,T253R (magenta) under far-red light conditions. The chromatography trace shows that the double mutation in the interface helix α8 quantitatively converts the dimer into a smaller species, without affecting the ground state of the chromophore that is kept in the FAD^OX state as indicated by the nearly equal absorbance at 370 nm and 450 nm for both wildtype and the double mutation (*Pd*LCry red, Abs$_{450nm}$/Abs$_{370nm}$ = 1.06; *Pd*LCry^E249R,T253R magenta, Abs$_{450nm}$/Abs$_{370nm}$ = 1.14). **b** Upon analytical SEC under continuous blue light conditions, wildtype *Pd*LCry and *Pd*LCry^E249R,T253R (dark and light blue) elute at the same volume, which coincides with the smaller species observed under far-red light conditions (light gray) for *Pd*LCry^E249R,T253R, supporting the constitutively monomeric state of *Pd*LCry^E249R,T253R. The dominant absorption at 370 nm compared to 450 nm (*Pd*LCry blue, Abs$_{450nm}$/Abs$_{370nm}$ = 0.28; *Pd*LCry^E249R,T253R cyan, Abs$_{450nm}$/Abs$_{370nm}$ = 0.28) highlights that the double mutation does not impact the ability of

*Pd*LCry to be converted into an FAD^ASQ-bound photoactivated state by prolonged illumination. **c, d** Normalized absorption spectra for sunlight photoreduction on ice show that **c** wildtype *Pd*LCry and **d** monomeric *Pd*LCry^E249R,T253R have similar sunlight photoreduction rates of t$_{1/2}$ 22.7 and 22.6 s. Arrows indicate absorbance changes at 370 nm, 404 nm and 450 nm upon photoreduction of dark-state FAD^OX (black) to the anionic FAD^ASQ radical (dark orange). **e, f** Absorption spectra for dark recovery at 18 °C after full (20 min) sunlight photoreduction show that compared to **e** wildtype *Pd*LCry, the **f** monomeric *Pd*LCry^E249R,T253R mutant has a faster dark recovery with an average t$_{1/2}$ of 157 s instead of 185 s. Arrows indicate absorbance changes at 370 nm, 404 nm and 450 nm upon dark recovery of the anionic FAD^ASQ radical (orange) to FAD^OX (black). The full spectra show a representative of three independent replicates. The t$_{1/2}$ values are the means of three independent measurements with standard error of the mean (SEM). See Supplementary Fig. 7b, c for average data plots.

within helix α8, and that single mutations such as E249R or T253R can be sufficient to disrupt the dimer (Fig. 6a). Moreover, we found that the unique Tyr266 of the PBL, which in other animal and animal-like CRYs is usually replaced by a phenylalanine, is important for stabilizing the dimer (Fig. 6c). The importance of both Glu249 and Tyr266 is further highlighted by our MD simulation of hydrogen bonds stabilizing the dimer interface (Supplementary Fig. 3f). The unique dimer architecture of *Pd*LCry places the PBL at a focal point connecting the chromophore, the dimer interface helix α8, and the CTT (Fig. 4c, d). The PBL has been identified by proteolysis experiments as a structural element affected by illumination of the chromophore in both type I[10] and type IV CRY[11] proteins. For the type IV CRY *Cl*Cry4, molecular dynamics simulations have confirmed that the PBL exhibits stronger mobility upon excitation of the chromophore[31]. In the context of our structural data, it is easy to imagine that a higher mobility of the PBL would affect the positioning of the dimer interface residue Tyr266, which is part of this loop. Furthermore, in the dark state, we observe that Pro274 and Gln271 of the PBL interact with Trp545 of the CTT, likely stabilizing this helix in its binding pocket (Fig. 4d). Indeed, our mutagenesis experiments strongly suggest that the PBL is functionally linked to both the dimer interface and the CTT (Fig. 6b, c). In addition, our *Pd*LCry structure reveals that the FFW motif of the CTT is not invariant in type I CRYs as previously hypothesized[14], with *Pd*LCry having a cysteine instead of a phenylalanine as the middle residue without affecting positioning or anchoring of the CTT (Fig. 4c, d).

Due to its role as a light-signal gatekeeper for the circalunar clock that controls the spawning behavior of *Platynereis dumerilii*[24,32], *Pd*LCry must be able to respond to dim light conditions. Indeed, in vitro experiments confirmed that *Pd*LCry, but not its ortholog *Dm*Cry, is photoreduced by a light source that mimics underwater moonlight illumination[24]. Our molecular structure suggests that, unlike *Dm*Cry, *Pd*LCry has an extended electron transport chain with Tyr352 as a fifth, highly surface-exposed, member (Fig. 4b). In the avian CRY protein *Cl*Cry4 the corresponding Tyr319 has been shown to confer an unusually high quantum yield to the protein[11]. We showed that Tyr352 also enhances the photoresponse of *Pd*LCry to naturalistic sunlight as its mutation leads to a less efficient FAD photoreduction under sunlight illumination (Supplementary Fig. 4a–c). This effect is potentially due to a lower quantum efficiency for FAD photoreduction in the mutants, consistent with a role of Tyr352 in electron transport to the FAD chromophore.

Interestingly, sunlight and moonlight illumination lead to different photoreduced states of *Pd*LCry in in vitro experiments[22], suggesting that the protein is able to distinguish between intensive and dim illumination as required for its in vivo activity[24]. Our current work confirms that *Pd*LCry can indeed form a stable dimer at dark conditions with the chromophore FAD in the resting oxidized state FAD^OX (Fig. 2). Based on the intense illumination conditions employed during our sample preparation, we hypothesize that the monomeric state we observe with fully reduced FAD^ASQ (Fig. 5)

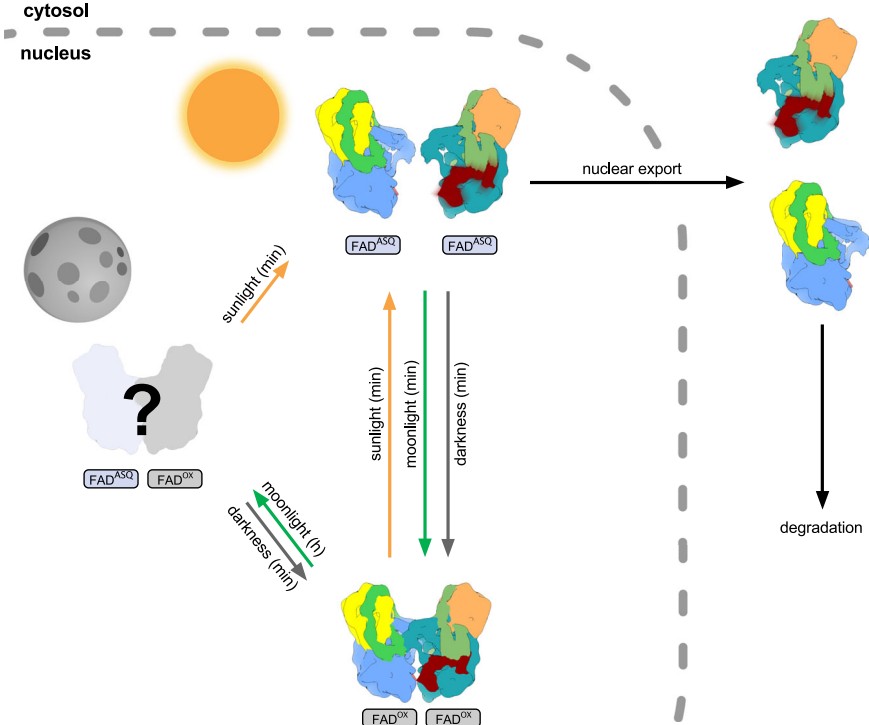

**Fig. 8 | The blue light stimulated dimer-to-monomer transition rationalizes previous in vitro and in vivo data.** Schematic overview of putative activation pathways at either low photon flux rates (moonlight) leading to a partial activation of the dimer within hours, or at high photon flux rates (sunlight) leading to a monomerization of the sample. In vitro assays have shown that the sunlight-activated state cannot be directly converted to the moonlight-activated state[22], but is only accessible via the dimeric dark state with FAD[OX] bound in both subunits. The monomeric FAD[ASQ]-bound state could be transported from the nucleus to the cytoplasm and subsequently targeted by a degradation pathway[24].

corresponds to the sunlight activated state rather than to the moonlight state.

In vitro experiments have shown that moonlight-mimicking low photon flux rates are only able to photoreduce half of the FAD present in a *Pd*LCry sample, even after hours of illumination, while higher flux rates lead to a full conversion of FAD within minutes[22]. However, it remains to be clarified whether the partially photoreduced moonlight-state consists of *Pd*LCry dimers with one oxidized and one photo-reduced FAD chromophore or of a mixture of oxidized and photo-reduced *Pd*LCry monomers. Our data suggests that structural elements linking the chromophore to the dimer interface, such as the PBL, could hypothetically allow the relay of information between both chromophores, inducing an asymmetry in *Pd*LCry. If thus activation of one chromophore leads to the dampening of the second chromo-phore, a stably half-activated moonlight state would be easy to reconcile at low photon flux rates. High flux rates however would overcome the dampening of the second chromophore, resulting in the formation of monomeric *Pd*LCry as observed in our experiments (Fig. 5). Such an asymmetric, activation-rate dependent input differ-entiation has also been observed in the rod photoactivation cascade, allowing the dimeric photoreceptor phosphodiesterase 6 (PDE6) to differentiate thermal noise from actual photon-detection[33].

Moreover, functional assays showed that while the moonlight-induced state can be converted to the sunlight-induced state, the sunlight-induced state cannot be directly converted to the moonlight-induced state[22]. Our finding that *Pd*LCry features an inverse photo-oligomerization mechanism rationalizes this observation: For conver-sion to the sunlight state, a partially activated moonlight state of *Pd*LCry, possibly an asymmetric dimer, simply requires higher photon flux rates to activate both subunits simultaneously, leading to the dissociation of the dimer in the sunlight state. Reassembly of the dimeric state, which we propose to be necessary for the transition from the monomeric sunlight state to the half-reduced moonlight state, may only be possible after FAD has returned to the resting state FAD[OX], available only by transition through the dark state (Fig. 8). Our proposed activation scheme would also explain the different dark recovery rates observed for wildtype *Pd*LCry and the monomeric mutant *Pd*LCry[E249R/T253R] in our time-resolved spectroscopy experi-ments (Fig. 7): while we monitored the sunlight state to dark state transition - thus from monomer to dimer - for the wildtype protein, the recovery measured for *Pd*LCry[E249R/T253R] was from the monomeric sun-light state to an artificial monomeric dark state.

The shift from a dimeric to a monomeric protein population only at high photon flux rates could also explain the different cellular localization observed for *Pd*LCry depending on the illumination conditions[24], assuming that only the monomer is targeted for nuclear export. Monomerization, nuclear export and subsequent cytosolic degradation could thus prevent erroneous activation of downstream signaling pathways by sunlight and allow input signal differentiation on the physiological level.

## Methods

### Sequence alignment and evolutionary analysis
The evolutionary analysis of the cryptochrome family was based on a maximum likelihood approach implemented in MEGA11 (version 11.0.10 build 211109)[34]. For this the evolutionary history was inferred by using the maximum likelihood method and the Jones-Taylor-Thornton (JTT) matrix-based model[35]. The bootstrap consensus tree inferred from 100 replicates[36] was taken to represent the evolutionary history of the taxa analyzed. Branches corresponding to partitions repro-duced in less than 50% bootstrap replicates were collapsed. Initial trees for the heuristic search were obtained automatically by applying Neighbour-Join (NJ) and BioNJ algorithms to a matrix of pairwise dis-tances estimated using the JTT model, and then selecting the topology

with superior log likelihood value. A discrete Gamma distribution was used to model evolutionary rate differences among sites (5 categories (+G, parameter = 0.8230)). This analysis involved 38 amino acid sequences, and there were a total of 1112 positions in the final dataset. For the list of used sequences see Supplementary Table 1.

Detailed pair-wise sequence alignments were calculated in Jalview (version 2.11)[37], relying on the JABAWS 2.2 web service[38] to run the Clustal Omega alignment method[39].

## Cloning, expression and purification of *Pd*LCry

Full-length N-terminally His6-tagged *Platynereis dumerilii Pd*LCry (GenBank UUF95169) was heterologously expressed in *Spodoptera frugiperda (Sf9*, Thermo Fisher Scientific Cat No. 11496015) insect cells using the Bac-to-Bac baculovirus expression system with the pCoofy27 expression vector, as described in Poehn et al.[22]. *Sf*9 cells were grown as suspension cultures in Sf-900 SFM III media (Thermo Fisher Scientific) at 27 °C, 90 rpm. 1 L of $1 \times 10^6$ *Sf*9 cells*mL$^{-1}$ were transfected with P1 virus stock and incubated at 27 °C for 72 h. Cells were harvested by centrifugation at $10,000 \times g$ for 20 min and stored at −80 °C until purification. All purification steps were carried out in dark or dim red light conditions. Columns were wrapped with aluminum foil to avoid light-activation of *Pd*LCry. The cell pellets were resuspended in lysis buffer (20 mM Tris pH 7.5, 150 mM NaCl, 20 mM imidazole, 5% glycerol, 5 mM β-mercaptoethanol) and lysed using a Branson sonifier (Analog Sonifier 450-CE). The lysate was centrifuged at $50,000 \times g$ for 45 min. The clarified supernatant was loaded onto a HisTrap FF crude 5 mL column (Cytiva), and *Pd*LCry was eluted in a 20 mM to 1000 mM imidazole gradient. Elution fractions containing *Pd*LCry were concentrated, diluted with no-salt buffer (50 mM Tris pH 7.5, 5% glycerol, 1 mM DTT) and loaded onto a 5 mL Hitrap Q HP anion exchange column (Cytiva). A gradient from 0% to 100% high salt buffer (50 mM Tris pH 7.5, 1 M NaCl, 5% glycerol, 1 mM DTT) was applied. *Pd*LCry containing fractions were pooled, concentrated and loaded onto a Superdex S200 10/300 GL or a HiLoad S200 16/60 SEC column (buffer 25 mM Bis-tris propane pH 8.0, 150 mM NaCl, 5% glycerol, 1 mM TCEP). Fractions containing pure *Pd*LCry were pooled, concentrated to 10 mg mL$^{-1}$ and snap frozen in liquid nitrogen for storage at −80 °C.

## Site-directed mutagenesis, expression, and purification of *Pd*LCry mutants

*Pd*LCry mutants were generated by QuikChange (Agilent) site-directed mutagenesis and verified by sequencing. The mutant constructs were expressed and purified essentially as described for wildtype *Pd*LCry.

## Cryo-EM sample preparation and data collection

All samples for cryo-EM were prepared in a dark lab under far-red light illumination (Osram OSLON SSL 120, GF CSSPM1.24). Concentrated samples of dark-adapted *Pd*LCry were diluted to a concentration of 0.7 mg mL$^{-1}$ using glycerol-free SEC buffer (25 mM Bis-tris propane pH 8.0, 150 mM NaCl, 1 mM TCEP). Dark-adapted samples were additionally supplemented with 0.01% fluorinated octyl maltoside (fOM). 4.5 μL of the mixture were then applied on either a glow-discharged UltrA-Ufoil R1.2/1.3 Au300 or a Quantifoil R1.2/1.3 Cu300 grid (Quantifoil Micro Tools GmbH), blotted for 3 sec and flash-frozen in liquid ethane using a Vitrobot Mark IV device (Thermo Fisher Scientific) set to 100% humidity at 21 °C. Grids were stored under liquid nitrogen conditions until usage. For blue light activation, grids were illuminated for 30 secs in the humidifier chamber of the freeze-plunger using a 455 nm blue light LED (Thorlabs M455L4, operated at 1000 mA). To ensure optimal illumination, a liquid-light guide (Thorlabs LLG03-4H) was used to bring the light source within 3 mm of the grid surface. Cryo-EM data was acquired using a Titan Krios G3i (Thermo Fisher Scientific) electron microscope operated at 300 kV. Images were collected automatically using EPU (version 2.12) (Thermo Fisher Scientific) on a Falcon III direct electron detector with a calibrated pixel size of 0.862 Å px$^{-1}$. For the dark state, movies were either acquired in integration mode using a total dose of 66 e$^-$ Å$^{-2}$ distributed among 19 frames, or in counting mode with a total dose of 27 e$^-$ Å$^{-2}$ distributed among either 24 or 48 frames. For the blue light state, movies were acquired in counting mode with a total dose of 30 e$^-$ Å$^{-2}$ distributed among 36 frames. Defocus values ranged from −0.3 to −2.0 μm.

## Image processing, quantification of monomer-to-dimer distribution, and model building

Image processing was performed using cryoSPARC (version 3.3.2+patch 220518)[40]. Movie stacks were first corrected for drift and beam-induction motion, and then used to determine defocus and other CTF-related values. Only high-quality micrographs with low drift metrics, low astigmatism and good agreement between experimental and calculated CTFs were further processed. On these high-quality micrographs putative particles were automatically picked based on expected protein diameter, extracted, and subjected to reference-free 2D classification. Based on this classification result, we estimated the number of particles showing either a monomer, a dimer or an ambiguous assembly - excluding classes showing clear artefacts. Representative 2D classes were then used for a template-based picking approach, particles extracted again, subjected to reference-free 2D classification to exclude artefacts, and subsequent 3D classification using C1 symmetry to identify high-quality particles. 3D classification was performed with a target resolution of 6 Å in order to ensure that secondary-structure elements are also included as a feature in the classification algorithm. Particles showing protein-like density features, especially features resembling secondary structure elements, were further refined using the non-uniform refinement strategy. For the dark state sample, C2 symmetry was enforced yielding a map at a global resolution of 2.6 Å. For the blue light-activated sample C1 refinement yielded a map at a nominal global resolution of 3.5 Å that was strongly anisotropic and thus filtered to 8 Å for further interpretation. For further details see Supplementary Fig. 2 and 5, and Supplementary Table 2.

The atomic model was built starting from the *Dm*Cry crystal structure (PDB ID 4JZY) and the *Pd*LCry amino acid sequence (GenBank UUF951691). First, Coot (version 0.9.4.7)[41] was used to semi-manually fit amino acids into a single subunit of the EM density map. Next, Phenix (version 1.20)[42] was used to generate a non-crystallographic symmetry (NCS)-related second subunit and both subunits were then simultaneously refined against the 3D density map. This process was iterated until the fit to the density map and geometric parameters converged. The final atomic model accounts for residues 31 to 556. For further details and statistics see Supplementary Table 2. Molecular visualization and analysis were done using UCSF ChimeraX (version 1.5)[43].

## MD simulations

The *Pd*LCry structure used for simulations was from this study. The structures for *Arabidopsis thaliana* Cry2 (*At*Cry2, PDB ID: 6K8I) and *Drosophila melanogaster* Cry (*Dm*Cry, PDB ID: 4JZY) were taken from the PDB database. The experimentally unresolved parts of *At*Cry2 were reconstructed using AlphaFold[44,45], where the *At*Cry2 sequence was used as an input. A root mean square deviation (RMSD) comparison of the backbone atoms between the AlphaFold generated *At*Cry2 monomeric structure and the PDB structure showed a difference of 1.8 Å.

Altogether 4 molecular dynamics (MD) simulations were performed using NAMD (version 2.14)[46–48] interfaced through the VIKING platform[49]. Two replica simulations of *Pd*LCry were carried out for 190 ns, while *At*Cry2 and *Dm*Cry were simulated once for 190 ns. All simulations initially included 10,000 conjugate gradient minimization steps. The simulations were then split into four distinct stages, first three used to equilibrate the system, and the fourth stage being used

for analysis (production simulation). The first three equilibration simulations were completed in an isothermal-isobaric (NPT) ensemble, while production simulations assumed a canonical (NVT) statistical ensemble. The three equilibration stages were carried out for 1 ns, 2 ns, and 2 ns, respectively, assuming different constraints in the system. In the first stage, the entire protein and FAD were harmonically constraint to the atomic positions taken from the experimental structure. In the second equilibration stage, the backbone atoms continued being constrained, while the side-chain atoms were free to move. The final equilibration stage continued with all atoms being free to move. Each production simulation was carried out over 190 ns with an integration time step of 2 fs, assuming the hydrogen atoms to be rigidly bonded to the corresponding heavier atoms. All simulations (equilibration and production) assumed a temperature of 310 K, controlled through the Langevin thermostat and an atmospheric pressure of 1 bar maintained through the Nosé–Hoover barostat. CHARMM36 force field with CMAP corrections was used in all simulations[50-55]. The investigated $Pd$LCry was set to be in the inactive state that is usually present in dark conditions, where the FAD cofactor is fully oxidized. Simulation parameters for the FAD cofactor were adopted from earlier studies[31,56-60]. NaCl was used to neutralize the total charge of the system and assumed at a concentration of 0.15 mol L$^{-1}$ in all simulations. Van der Waals and electrostatic interactions were treated with the cut-off distances of 12 Å, while the particle-mesh Ewald (PME) summation method was used to treat the long-range electrostatic interactions[61]. The size of the simulation box for $At$Cry2 was 102 Å × 136 Å × 140 Å and contained 201,877 atoms, for $Dm$Cry the simulation box was 134 Å × 105 Å × 135Å with 199,362 atoms, and for $Pd$LCry the simulation box was 111 Å × 118 Å × 109 Å, containing 148,535 atoms.

Convergence was assessed by following the RMSD evolution over time, with the RMSD defined as follows:

$$RMSD(t) = \sqrt{\frac{1}{N}\sum_{i=1}^{N}\left|\vec{r}_i(t) - \vec{r}_i^{ref}\right|^2} \qquad (1)$$

where $N$ is the number of backbone atoms in a given dimer, $\vec{r}_i(t)$ is the position vector of the $i$th atom at a time instance, $t$, and $\vec{r}_i^{ref}$ is the position vector of the $i$th atom at the reference time point. To analyze the buried area of the dimeric cryptochrome structures, the difference between the solvent accessible surface area (SASA) was computed as:

$$S_0 = (S_1 + S_2) - S_d \qquad (2)$$

Here $S_0$ is the interaction surface (or the buried area) between the monomers in a dimer, $S_1$ and $S_2$ are the SASAs of the two monomers and $S_d$ is the SASA value of the whole dimer. The SASA maximal speed molecular surfaces (MSMS) algorithm[62] in VMD[63] was used to compute the values in Eq. (1), assuming each atom having a radius of 1.4 Å. Hydrogen bond analysis was performed using the HBonds plugin in VMD (version 1.9.3)[63].

## Analytical SEC and inline UV/Vis spectroscopy

For inline UV/Vis spectroscopy analytical SEC runs were performed in a dark lab under far-red light illumination (Osram OSLON SSL 120, GF CSSPM1.24) using a Superdex 200 Increase 5/150 GL (Cytiva) SEC column connected to an bioinert AZURA HPLC system (Knauer) equipped with an autosampler and a TIR multiwavelength detector set to measure absorption at 280 nm, 370 nm, 404 nm and 450 nm. Data collection was controlled using the PurityChrom (version 5.09.115) software. The flow rate for all runs was 0.16 mL min$^{-1}$. Concentrated samples of dark-adapted $Pd$LCry were diluted to a concentration of 5 mg mL$^{-1}$ using a buffer containing 25 mM Bis-tris propane (pH 8.0), 150 mM NaCl, and 1 mM TCEP, prior to injection of 7 μL per run. For dark condition runs, the chromatography column was wrapped in

aluminum foil. For blue light condition runs, samples were pre-activated for 60 sec using a 455 nm blue light LED (Thorlabs M455L4, operated at 1000 mA) at a distance of 1 cm. During the run, the chromatography column was continuously illuminated by a 451 nm blue light LED array (24x Osram OSLON SSL 120, GF CSSPM1.14) installed at a distance of 10 cm in parallel to the column and reflected by mirror elements.

Analytical SEC runs without inline UV/Vis spectroscopy were carried out under far-red light conditions using a Superdex 200 10/300 GL column wrapped in aluminum foil connected to a Biorad NGC Quest10 Plus Chromatography System. For each SEC run, 100 μL sample diluted to a concentration of 2 mg mL$^{-1}$ concentration was injected. Data collection was controlled using the ChromLab (version 6.1) software.

## UV/Vis spectroscopic analyses of blue light and sunlight photoreduction and dark recovery

UV/Visible absorption spectra of the purified $Pd$LCry proteins in final SEC purification buffer (25 mM Bis-tris propane pH 8.0, 150 mM NaCl, 5% glycerol, 1 mM TCEP), supplemented with 1 mM DTT, were recorded on a Tecan Spark 20 M plate reader using the Spark Control (version 2.2) software. A light state spectrum of $Pd$LCry with fully photoreduced FAD$^{ASQ}$ was collected after illuminating dark-adapted $Pd$LCry for 2 min with a 450 nm blue LED (41 W m$^{-2}$ at sample). To analyze sunlight dependent FAD photoreduction kinetics, dark-adapted $Pd$LCry was illuminated with naturalistic sunlight (Marine Breeding Systems GmbH[22]) with 4.6 W m$^{-2}$ intensity at the sample for up to 20 min (protein kept on ice) and complete UV–VIS spectra (300–700 nm) were collected after different time intervals. Sunlight photoreduction kinetics (on ice) were determined based on changes of 450 nm absorbance values, that were extracted from these UV/VIS spectra after normalization to their isosbestic point at 416 nm. Normalization of sunlight photoreduction experiments was required to correct for baseline shifts caused by sample handling at each measurement time point, and (in the case of kinetic studies) because two additional samples had to be measured to determine FAD photoreduction after 10 s and 20 s sunlight illumination. Dark recovery kinetics (FAD reoxidation) after photoreduction by 20 min sunlight illumination were determined at 18 °C by recording absorbance changes at 450 nm.

Absorption spectra and kinetics were analyzed using GraphPad PRISM (version 9.5.1-733). The half-life time values ($t_{1/2}$ [s]) for dark recovery and photoreduction kinetics were calculated by fitting a single exponential curve to the experimental data. The single exponential model is described as $Y = (Y_0 - Y_{plateau})^{(-KX)} + Y_{plateau}$, where $Y_0$ and $Y_{plateau}$ correspond to the 450 nm absorbance values at the start- and end-points of the measurement, K represents the rate constant [1 s$^{-1}$] and the half-life time $t_{1/2} = \ln(2)*K^{-1}$ [s]. The standard error of the mean (SEM) was used to describe the error of the measurements and of the $t_{1/2}$ values. $SEM = \frac{SD}{\sqrt{n}}$ where SD is the standard deviation and $n$ is the number of replicates.

## Reporting summary

Further information on research design is available in the Nature Portfolio Reporting Summary linked to this article.

# Data availability

The EM maps for the dark state and blue light activated state have been deposited in the EMDB under accession codes EMD-17429 and EMD-17553. The raw electron microscopy data are available on request due to their large size. Atomic coordinates for dark state $Pd$LCry have been deposited in the Protein Data Bank under the accession code PDB 8P4X. The sequence analysis data, SEC data, MD data, and UV/Vis data have been deposited in the Zenodo repository under the https://doi.org/10.5281/zenodo.8420035.

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

## Acknowledgements

The authors thank Christoph Klatt for contributions at an early stage of the project, and acknowledge use of the ESRF beamline CM01 during an early stage of the project. We are grateful for laboratory support by Alexandra Schneider, and by the IMB Media Lab, Protein Production, and Proteomics Core Facilities. We acknowledge access to the cryo-EM infrastructure of StruBiTEM (Cologne, funded by DFG Grant INST 216/949-1 FUGG), to the computing infrastructure of CARL (Oldenburg, supported by the DFG and the Ministry for Science and Culture of Lower Saxony), and to the computing infrastructure of CHEOPS (Cologne, funded by DFG Grant INST 216/512/1 FUGG). Additional computing infrastructure was provided by the Resource Allocation Board on the supercomputer Lise and Emmy at NHR@ZIB and NHR@Göttingen under project nip00058 (to I.A.S.) and a hardware grant from NVIDIA (to E.B.). This study was supported by funding from the DFG (SFB1372-Sig03 to E.B; SFB1372-Sig05 to I.A.S.; GRK1885 to I.A.S.; TRR386/1-2023 HYP*-MOL, no 514664767 to I.A.S.); a DFG fellowship of the Excellence Initiative by the Graduate School Materials Science in Mainz (GSC 266, to S.K.), the Ministry for Science and Culture of Lower Saxony (SMART and DyNano to I.A.S.), and the Volkswagen Foundation (Lichtenberg Professorship to I.A.S.).

## Author contributions

H.H.V., S.K., and C.H. prepared protein samples; H.H.V. carried out protein mutagenesis; H.H.V., H.B., D.D., and C.H. carried out SEC experiments; H.H.V. carried out time-resolved spectroscopy experiments; H.B. carried out sequence analysis; H.B., M.G., and G.J. prepared cryo-EM samples; M.G. and G.J. acquired cryo-EM data; H.B. and E.B. performed cryo-EM data analysis; H.B. prepared atomic models with input from E.W. and E.B.; M.H. carried out MD experiments; H.H.V., H.B., M.H., I.A.S., E.W., and E.B. interpreted data; E.B. and E.W. conceived the project; I.A.S., E.W., and E.B. supervised the project; E.B. wrote the manuscript with input from all authors.

## Funding

## Competing interests

The authors declare no competing interests.
