## [Peer Review File · Nature Communications]

A marine cryptochrome with an inverse photo-oligomerization mechanismREVIEWER COMMENTS

Reviewer #1 (Remarks to the Author):

The manuscript presents an unusual photoactivation mechanism for a new marine cryptochrome, which has previously not been detected. The proposed inverse photo-oligomerization mechanism is strongly supported by biochemical and structural characterization, and the key elements of the activation cascade are well demonstrated and confirmed by functional studies on mutant sequences. The results are convincingly interpreted on a cellular and physiological level. These findings advance our understanding on how cryptochromes function, and thus advance the field significantly.

I strongly support the publication of this study in Nature Communications in its current state. The paper is well written, the data supports the conclusions drawn and the discussion is well-balanced with respect to the existing literature. I have a couple of minor comments that could be addressed.

1: Line 303-304: this sentence reads like it suggests that only E249 and Thr253 participate in the dimerization, it might be worth rephrasing.

2: Line 325-327: while the positioning of Tyr352 strongly supports the theory that it is a part of the electron transport chain, more evidence is needed to state this as a fact. Functional studies on a Tyr352 mutant could be conducted for additional proof.

3: Line 333-335: is there any indication that both chromophores need to convert for the monomerization to occur?

4: While it is suggested from the text, it is not confirmed that the blob picking parameters (the minimum and maximum particle diameter) and the extraction box size matched and allowed both oligomerization states to be observed for the two datasets. As one of the main results supporting the light-dependent oligomerization is the results of 2D classification, it is important to confirm that there was no bias when obtaining these results.

5: Supplementary figures 2 and 5: the images showing ab-initio reconstructed volumes show unusually high resolution. Was the maximum resolution parameter changed when running this job?

Reviewer #2 (Remarks to the Author):

PdLCRY is essential for a marine bristle worm to distinguish sunlight and moonlight, but the molecular mechanism remains unknown. Vu et al. found that PdLCRY forms a dimer without blue-light but the dimer dissociates into monomers in the presence of blue-light. Using cryo-EM, the authors identified the dimer interfaces that are essential for PdLCRY dimer assembly. The structural work of the dimer is well-carried and the dimer map is of high quality. I believe this manuscript will surely be the first of many papers that will eventually reveal the mechanisms of PdLCRY. I have no major issues but a few minor issues, which might be helpful for the authors to improve the manuscript:

1. The author needs to provide more details about how to quantify the dimer-monomer distribution in Fig. 2b and Fig. 5b. Are these numbers from 2D classifications or 3D classifications? Are there any particles discarded before the quantification?
2. It would be nice to have the 2D class averages in Fig. 2b and Fig. 5b on the same scale, just for fair visual comparisons.
3. I noticed that the authors have collected a large fraction of data using integration mode on falcon III, which is not common in the field. I am wondering if the authors have tried to do reconstructions on integration mode data alone. How about the resolution in that case?
4. The raw micrographs should be included in supplementary Fig. 2 and 5.
5. In supplementary Fig. 2 and 5, please box the 3D classes used for subsequent processing.
6. I would suggest toning down the interpretation of the monomeric map in Fig. 5c-d. The current map quality seems not sufficient to support the conclusion about conformational changes in lines 221-226.
7. I am not sure if this experiment is straightforward: is it possible to disrupt the dimer interface using mutations and look at the in vivo phenotype of the animals?

Reviewer #3 (Remarks to the Author):

In this work, Vu et al. study the light-responsive cryptochrome PdLCry with Cryo-EM, time-resolved spectroscopy and MD simulations. They solve the structure of both the dark state and the light-activated state and use MD simulations to characterize the dimer interface. The authors put forward a mechanism by which the cryptochrome may have two different light-activated forms if activation occurs by very dim light (moonlight) or by bright light.

This work takes advantage of multiple experimental and computational techniques to uncover the activation mechanism of PdLCry. The manuscript is generally clear and well written. I only have some

comments on the MD part, which is seemingly a minor part of the work but substantiates some of the conclusions.

1) There are some methodological details missing regarding the MD simulations. The authors state "The simulations were then split into four distinct stages, first three used to equilibrate the system, and the fourth stage being used for analysis". It is not clear (a) how the system was heated to the target temperature, (b) which ensemble was considered during equilibration and production, (c) how long were these four stages, and if the 190 ns comprise only the production part. A Supplementary Methods section should be added that explains all these details.

2) The MD simulations were not characterized except for the few analyses shown in the Supplementary Information. An assessment of convergence/equilibration is mandatory for MD simulations. Confidence intervals for the SASA average values should be reported.

3) The authors computed the SASA of the entire dimer and of the two monomers in order to analyze the buried area. They also state that ". Restricted selections were used in the calculations to avoid protein pockets affecting the results", which is a vague statement. Which selections were performed exactly? I do not think that pockets inside either monomer would affect the $S0$ value, as they should get cancelled in the difference. Maybe I am missing something, and I urge the authors to clarify this point, possibly in the Supplementary Information.

4) The authors enforced C2 symmetry in the CryoEM reconstruction stages, which seems to be supported by the fact that no obvious asymmetry was present. Is the symmetry preserved (apart from minor fluctuations) also in the MD simulations? This would substantiate the choice of enforcing symmetry during CryoEM reconstruction.

Point-by-point response to the REVIEWER COMMENTS

First of all, we thank all three referees for their time taken to critically evaluate our manuscript, and for their thoughtful and constructive comments that helped us to improve our manuscript. We have strived to address all raised concerns in our revised manuscript and will respond to all comments below, which are highlighted in blue italics.

Reviewer #1 (Remarks to the Author):

The manuscript presents an unusual photoactivation mechanism for a new marine cryptochrome, which has previously not been detected. The proposed inverse photo-oligomerization mechanism is strongly supported by biochemical and structural characterization, and the key elements of the activation cascade are well demonstrated and confirmed by functional studies on mutant sequences. The results are convincingly interpreted on a cellular and physiological level. These findings advance our understanding on how cryptochromes function, and thus advance the field significantly.

I strongly support the publication of this study in Nature Communications in its current state. The paper is well written, the data supports the conclusions drawn and the discussion is well-balanced with respect to the existing literature. I have a couple of minor comments that could be addressed.

1: Line 303-304: this sentence reads like it suggests that only E249 and Thr253 participate in the dimerization, it might be worth rephrasing.

We thank the reviewer for pointing out the impreciseness of this sentence and have rephrased it as follows:

“For PdLCry, we could show that dimerization depends on the presence of specific amino acids within helix $\alpha 8$, and that single mutations such as E249R or T253R can be sufficient to disrupt the dimer (Fig. 6a).”

2: Line 325-327: while the positioning of Tyr352 strongly supports the theory that it is a part of the electron transport chain, more evidence is needed to state this as a fact. Functional studies on a Try352 mutant could be conducted for additional proof.

We thank the reviewer for pointing out this experimental idea, which we have now carried out. In the revised manuscript, we include data on the photoreduction of the Tyr352 mutants PdLCry^{Y352A} and PdLCry^{Y352D} in Supplementary Figure 4. Consistent with our hypothesized functional role of Tyr352 in the electron transport chain, both PdLCry^{Y352A} and PdLCry^{Y352D} proteins show a slower response to naturalistic sunlight and do not completely photoreduce their FAD cofactor after 20 min sunlight illumination, whereas in wildtype PdLCry FAD is almost completely photoreduced after 5 to 20 min sunlight illumination. SEC confirms that this is not due to a change in the oligomeric state of the dark state, as both mutants remain dimeric in the dark. This control data we now show in Supplementary Figure 6.

In the manuscript we have added the following to the results section

*“To test whether this terminal tyrosine does indeed affect the efficiency of FAD photoreduction in PdLCry, we replaced Tyr352 with either an alanine or an aspartate as found in DmCry and CraCry (Fig. 4b, Supplementary Fig. 1). Both PdLCry^{Y352A} and PdLCry^{Y352D} form dimers in the dark (Supplementary Fig. 6i, j), but both mutations lead to a less efficient FAD photoreduction under illumination conditions mimicking naturalistic sunlight conditions at the habitat of *P. dumerilii*²². Indeed, for both mutations photoreduction was slower and incomplete after 20 min sunlight illumination (Supplementary Fig 4a, b), different from wildtype PdLCry that is almost completely photoreduced using identical conditions (Supplementary Fig. 4c).*

And the following to the discussion section

"We showed that Tyr352 also enhances the photoresponse of PdLCry to naturalistic sunlight as its mutation leads to a less efficient FAD photoreduction under sunlight illumination (Supplementary Fig. 4a-c). This effect is potentially due to a lower quantum efficiency for FAD photoreduction in the mutants, consistent with a role of Tyr352 in electron transport to the FAD chromophore."

3: Line 333-335: is there any indication that both chromophores need to convert for the monomerization to occur?

We thank the reviewer for highlighting that we were not clear enough with our statement. In the current work, we focused on the dark state and a fully photoreduced state at saturating light conditions to ensure that we form homogeneous and unambiguous biochemical states for our cryo-EM experiments.

Therefore, we do not know whether or not the activation of a single subunit, such as likely occurring during moonlight stimulation (see Pöhn et al. 2022), will disrupt the dimer or not. Therefore, we refrained from making any statements on the moonlight state in this work. We have now rephrased this section as following in order to make sure that this remaining uncertainty becomes obvious to the reader:

"Our current work confirms that PdLCry can indeed form a stable dimer at dark conditions with the chromophore FAD in the resting oxidized state FAD^{OX} (Fig. 2). Based on the intense illumination conditions employed during our sample preparation, we hypothesize that the monomeric state we observe with fully reduced FAD^{ASQ} (Fig. 5) corresponds to the sunlight activated state rather than to the moonlight state.

In vitro experiments have shown that moonlight-mimicking low photon flux rates are only able to photoreduce half of the FAD present in a PdLCry sample, even after hours of illumination, while higher flux rates lead to a full conversion of FAD within minutes²². However, it remains to be clarified whether the partially photoreduced moonlight-state consists of PdLCry dimers with one oxidized and one photoreduced FAD chromophore or of a mixture of oxidized and photoreduced PdLCry monomers."

4: While it is suggested from the text, it is not confirmed that the blob picking parameters (the minimum and maximum particle diameter) and the extraction box size matched and allowed both oligomerization states to be observed for the two datasets. As one of the main results supporting the light-dependent oligomerization is the results of 2D classification, it is important to confirm that there was no bias when obtaining these results.

Indeed, we took great care to use identical parameters for picking and classifying particles in both datasets in order to avoid introducing any bias. We have highlighted this briefly in the main text as follows

"Data acquisition and unsupervised 2D classification using identical conditions as for the dark state dataset indeed revealed almost exclusively monomeric particles (Fig. 5b)."

and extended the Material and Methods section as described in our answer to comment #1 of reviewer #2.

5: Supplementary figures 2 and 5: the images showing ab-initio reconstructed volumes show unusually high resolution. Was the maximum resolution parameter changed when running this job?

We extensively screened for optimal conditions during unsupervised 3D classification and found that ab-initio reconstructions targeted at SSE resolution (6 Å as the high-resolution limit) outperforms all other

tested sorting strategies in our hands. This is in agreement with the recommendations for working with small membrane proteins in cryoSPARC (<https://guide.cryosparc.com/processing-data/tutorials-and-case-studies/tutorial-tips-for-membrane-protein-structures>). To highlight this to the reader – and hopefully allow them similar success with their own protein of interest – we have added a short sentence to the Material and Methods section as follows:

“3D classification was performed with a target resolution of 6 Å in order to ensure that secondary-structure-elements are also included as a feature in the classification algorithm.”

Reviewer #2 (Remarks to the Author):

PdLCRY is essential for a marine bristle worm to distinguish sunlight and moonlight, but the molecular mechanism remains unknown. Vu et al. found that PdLCRY forms a dimer without blue-light but the dimer dissociates into monomers in the presence of blue-light. Using cryo-EM, the authors identified the dimer interfaces that are essential for PdLCRY dimer assembly. The structural work of the dimer is well-carried and the dimer map is of high quality. I believe this manuscript will surely be the first of many papers that will eventually reveal the mechanisms of PdLCRY. I have no major issues but a few minor issues, which might be helpful for the authors to improve the manuscript:

1. The author needs to provide more details about how to quantify the dimer-monomer distribution in Fig. 2b and Fig. 5b. Are these numbers from 2D classifications or 3D classifications? Are there any particles discarded before the quantification?

We thank the reviewer for highlighting that our description of the processing leading to the quantification of the dimer to monomer distribution was lacking critical information required for understanding our approach. We are aware that quantification of oligomeric states using cryo-EM is error prone and distributions are easily affected by various parameters of the processing pipeline. Therefore, we refrained from stating exact numbers in our manuscript but tried to rather put emphasis on the almost opposite behavior of dark and blue light states: while the former shows mainly dimers with few monomers, the latter is almost exclusively monomeric.

Nevertheless, we tried to minimize user bias by performing the quantification of monomer to dimer distribution on the stage of blob-picker selected particles – thus at a stage where the user input was limited to specifying a size range expected for the particles. Importantly, we used the same parameters for the blob-picker for both datasets at this stage – which indeed also identified monomers as can be seen in the dark state 2D classification results. However, not all user bias can be avoided in this process as 2D class averages were assigned as either “monomer”, “dimer” or “ambiguous” classes by hand. Classes showing clear artefacts such as ethane contaminations were excluded prior to this.

We have now expanded the corresponding section in the Materials and Methods section as follows:

“On these high-quality micrographs putative particles were automatically picked based on expected protein diameter, extracted, and subjected to reference-free 2D classification. Based on this classification result, we estimated the number of particles showing either a monomer, a dimer or an ambiguous assembly – excluding classes showing clear artefacts. Representative 2D classes were then used for a template-based picking approach, particles extracted again, subjected to reference-free 2D classification to exclude artefacts, and subsequent 3D classification using C1 symmetry to identify high-quality particles.”

(see also our answer to comment #4 of reviewer #1 who was commenting on the same issue)

2. It would be nice to have the 2D class averages in Fig. 2b and Fig. 5b on the same scale, just for fair visual comparisons.

In our original figures uploaded, the 2D class averages of the dark state and the blue light state are on the same scale. We thank the reviewer for making us aware that this might have been changed during the automatic assembly of the pdf, and will make sure that this does not happen again. For immediate comparison, we have assembled both panels from Fig 2 and 5 into point-by-point Figure 1:

Point-by-point Figure 1: a Representative 2D class averages of an unsupervised 2D classification of the dark state PdLCry cryo-EM sample. Reproduced from Fig. 2b. b Representative 2D class averages of an unsupervised 2D classification of the photoactivated PdLCry cryo-EM sample. Reproduced from Fig. 5b.

3. I noticed that the authors have collected a large fraction of data using integration mode on falcon III, which is not common in the field. I am wondering if the authors have tried to do reconstructions on integration mode data alone. How about the resolution in that case?

We agree with the reviewer that the general view in field is that counting mode will outperform the integration mode of the Falcon III camera. For our microscopy setup initial test datasets however showed that – when normalized for beam time, since integration mode acquisition has a significantly higher throughput– both camera settings can yield reconstructions reaching similar final resolutions.

Since PdLCry was one of the first real samples acquired in our facility after these test datasets, we still used a variety of acquisition settings as shown in Supplementary Figure 2. All four datasets gave similar resolution reconstructions if processed individually, however with the UltraUfoil grid datasets yielding more good particles per micrograph compared to the copper grid. As we could boost the data quality (not so much the FSC resolution) by combining all available datasets, we choose to deposit this combined structure to the EMDB and use it for model building.

4. The raw micrographs should be included in supplementary Fig. 2 and 5.

We have added a representative raw, motion corrected micrograph to Supplementary Figures 2 and 5.

5. In supplementary Fig. 2 and 5, please box the 3D classes used for subsequent processing.

In addition to the green highlight, we have now also boxed the 3D classes used for subsequent processing in Supplementary Figures 2 and 5 to make sure this is also visible if printed in greyscale.

6. I would suggest toning down the interpretation of the monomeric map in Fig. 5c-d. The current map quality seems not sufficient to support the conclusion about conformational changes in lines 221-226.

We agree that our original wording of this section lends itself to be read as a solid conclusion supported by our structural data, and not as a hypothesis as we intended it to be. We have therefore shortened this part considerably, keeping only the actual description of the model fit together with the reference to the previously observed mobility of helix 22 in CraCry and the CTT in DmCry:

"Nevertheless, rigid body fitting of a single subunit from our dark state atomic model into the monomeric 3D map reveals a general fit (Fig. 5c), however with poorer fits for helix α 22 and the CTT (Fig. 5d). Both elements have previously been suggested to become more mobile upon blue light activation for other CRY proteins^{13,21}.

7. I am not sure if this experiment is straightforward: is it possible to disrupt the dimer interface using mutations and look at the in vivo phenotype of the animals?

We agree that it would be very interesting to look at the in vivo phenotypes of the mutations we have studied in vitro, but this is clearly beyond the scope of this manuscript revision:

P. dumerilii worms have a generation time of about 3 to 4 months, so homocygote worms could be obtained after about 9 months at best, more realistically after about 1 year. After successful generation of mutant worms, behavioral experiments will take additional months, as data must be collected for several lunar cycles to obtain reliable results. Depending on the outcome of these experiments, additional time-consuming assays, such as immunohistochemical analysis to identify the cellular localization of mutant PdLCry, may be required before final conclusions can be drawn.

Reviewer #3 (Remarks to the Author):

In this work, Vu et al. study the light-responsive cryptochrome PdLCry with Cryo-EM, time-resolved spectroscopy and MD simulations. They solve the structure of both the dark state and the light-activated state and use MD simulations to characterize the dimer interface. The authors put forward a mechanism by which the cryptochrome may have two different light-activated forms if activation occurs by very dim light (moonlight) or by bright light.

This work takes advantage of multiple experimental and computational techniques to uncover the activation mechanism of PdLCry. The manuscript is generally clear and well written. I only have some comments on the MD part, which is seemingly a minor part of the work but substantiates some of the conclusions.

1) There are some methodological details missing regarding the MD simulations. The authors state "The simulations were then split into four distinct stages, first three used to equilibrate the system, and the fourth stage being used for analysis". It is not clear (a) how the system was heated to the target temperature, (b) which ensemble was considered during equilibration and production, (c) how long were these four stages, and if the 190 ns comprise only the production part. A Supplementary Methods section should be added that explains all these details.

We thank the reviewer for highlighting that our original description was lacking methodological details. We have now added further details to the Materials and Methods section as follows:

"The simulations were then split into four distinct stages, first three used to equilibrate the system, and the fourth stage being used for analysis (production simulation). The first three equilibration simulations were completed in an isothermal-isobaric (NPT) ensemble, while production simulations assumed a canonical (NVT) statistical ensemble. The three equilibration stages were carried out for 1 ns, 2 ns and 2 ns, respectively, assuming different constraints in the system. In

the first stage the entire protein and FAD were harmonically constraint to the atomic positions taken from the experimental structure. In the second equilibration stage the backbone atoms continued being constrained, while the side-chain atoms were free to move. The final equilibration stage continued with all atoms being free to move. Each production simulation was carried out over 190 ns with an integration time step of 2 fs, assuming the hydrogen atoms to be rigidly bonded to the corresponding heavier atoms. All simulations (equilibration and production) assumed a temperature of 310 K, controlled through the Langevin thermostat and an atmospheric pressure of 1 bar maintained through the Nosé–Hoover barostat.”

2) The MD simulations were not characterized except for the few analyses shown in the Supplementary Information. An assessment of convergence/equilibration is mandatory for MD simulations. Confidence intervals for the SASA average values should be reported.

Following the request of the reviewer, we have added an additional panel with the time evolution of the root-mean-square-displacement of the protein backbone to Supplementary Figure 3. The following additional explanations have been added to the figure legend:

“c Evolution of the root mean square deviation (RMSD) of the dimeric backbone atoms from the reference configuration taken after the equilibrium simulations. The overall RMSD for the studied dimers converges to the values below 3 Å for the AtCry2 (green trace) and PdLCry simulation #2 (light blue trace) which indicate stable dimers. RMSD for the crystal DmCry dimer simulation (orange trace) shows several peaks, which appear due to the slight relative oscillatory motion of the two monomers in the DmCry dimer structure. A slight increase of the RMSD in the case of the PdLCry simulation #1 (dark blue trace) is also caused by relative motion of the two monomers, which remarkably was not seen in the replica PdLCry simulation #2. This motion is reflected in the SASA time evolution shown in d, which demonstrates that in the case of PdLCry simulation #1 the buried SASA value drops shortly after the beginning of the simulation, and then increases again to the characteristic value also seen in the case of PdLCry simulation #2. Note that the RMSD peculiarities in the case of DmCry and PdLCry simulation #2, however, are rather minor, as overall the RMSD stays well below 4 Å, indicating overall stable structures, as RMSD values for stable single proteins of the size of cryptochrome are expected to be within a 2-4 Å range“

Following the second comment of the reviewer regarding the confidence intervals for the SASA average values, we have added an additional panel to Supplementary Figure 3 where we use the violin-representation to show the requested statistical data. The following explanation was added to the figure legend:

“e Violin plot representation of the calculated SASA values. The figure shows the mean values and indicates the corresponding 25-75% percentile intervals of the data. The skewed distribution for PdLCry simulation #1 is due to the significantly lower SASA values at the beginning of the simulation compared to the results obtained after 190 ns. The average SASA values of the PdLCry and the AtCry2 structures and their confidence intervals are well separated from the result obtained from the DmCry simulation.”

3) The authors computed the SASA of the entire dimer and of the two monomers in order to analyze the buried area. They also state that ". Restricted selections were used in the calculations to avoid protein pockets affecting the results", which is a vague statement. Which selections were performed exactly? I do not think that pockets inside either monomer would affect the S0 value, as they should get cancelled in the difference. Maybe I am missing something, and I urge the authors to clarify this point, possibly in the Supplementary Information.

The reviewer is correct that the mentioned statement is indeed misleading. It was intended to refer to some technicalities of how the SASA in VMD was measured but does not add any important information. Moreover, we agree with the reviewer that the SASA contributions from the pockets would indeed cancel

each other once we calculate the S0 value. To avoid confusions, we have removed the problematic sentence from the revised manuscript.

4) The authors enforced C2 symmetry in the CryoEM reconstruction stages, which seems to be supported by the fact that no obvious asymmetry was present. Is the symmetry preserved (apart from minor fluctuations) also in the MD simulations? This would substantiate the choice of enforcing symmetry during CryoEM reconstruction.

This question is not easy to answer. As confirmed by the low RMSD values (see answer #2), we did not observe any gross deviations from C2 symmetry during the MD simulations. However, we did observe a slight relative oscillatory motion of the two monomers in both the DmCry simulation and one of the two replicates of the PdLCry simulations, which complicates a quantitative analysis of the symmetry in the MD simulations apart from a qualitative statement.

In this regard, we would like to emphasize that our 3D sorting procedure was performed without using symmetry (specifying C1 symmetry) and yielded classes with two symmetrically arranged subunits (see Supplementary Figure 2). Furthermore, incorrectly applying C2 symmetry would have obscured the side chain density in our final reconstruction, which was not the case. In addition, we also performed a reconstruction using C1 symmetry (Point-by-point Figure 2a), which did not show any obvious differences between the two subunits when compared visually (Point-by-point Figure 2b). A difference map also shows no interpretable differences (Point-by-point Figure 2c).

Point-by-point Figure 2: **a** Density map of a C1-symmetric reconstruction of the dark state dataset. The masked and corrected resolution is 3.3 Å. **b** To investigate changes at the side-chain level the map was low-pass filtered by dampening the high frequencies with a B-factor of 100 Å² to remove noise as implemented in ChimeraX. Next, the density regions belonging to either subunit #1 or #2 were separated, and the subunit #2 map fitted onto the subunit #1 map (yellow and blue). The pdb model is shown in white for reference. **c** Positive (green) and negative (red) difference maps of the maps shown in panel b also show no interpretable differences between the parts of the map corresponding to subunit #1 and #2 in the C1-symmetric reconstruction.

REVIEWERS' COMMENTS

Reviewer #1 (Remarks to the Author):

We greatly appreciate the authors' thorough consideration of our comments. With these revisions, all our concerns have been thoughtfully addressed. I am now happy to recommend this paper for publication.

Reviewer #2 (Remarks to the Author):

The authors have answered all of my questions.

I have no more concerns.

Reviewer #3 (Remarks to the Author):

The authors have satisfactorily addressed my comments. I therefore fully recommend publication of this paper.

Point-by-point response to the REVIEWER COMMENTS

We again thank the reviewers for their time taken to critically evaluate our manuscript, and are happy that we could address all concerns.

Reviewer #1 (Remarks to the Author):

We greatly appreciate the authors' thorough consideration of our comments. With these revisions, all our concerns have been thoughtfully addressed. I am now happy to recommend this paper for publication.

Reviewer #2 (Remarks to the Author):

The authors have answered all of my questions.
I have no more concerns.

Reviewer #3 (Remarks to the Author):

The authors have satisfactorily addressed my comments. I therefore fully recommend publication of this paper.